# POLICY-BASED SELF-COMPETITION FOR PLANNING PROBLEMS

**Jonathan Pirnay**[1,2]**, Quirin Göttl**[1]**, Jakob Burger**[1] **& Dominik G. Grimm**[1,2]
[1]Technical University of Munich, Campus Straubing for Biotechnology and Sustainability
[2]Weihenstephan-Triesdorf University of Applied Sciences
`{jonathan.pirnay,dominik.grimm}@{hswt.de,tum.de}`

## ABSTRACT

AlphaZero-type algorithms may stop improving on single-player tasks in case the value network guiding the tree search is unable to approximate the outcome of an episode sufficiently well. One technique to address this problem is transforming the single-player task through self-competition. The main idea is to compute a scalar baseline from the agent's historical performances and to reshape an episode's reward into a binary output, indicating whether the baseline has been exceeded or not. However, this baseline only carries limited information for the agent about strategies how to improve. We leverage the idea of self-competition and directly incorporate a historical policy into the planning process instead of its scalar performance. Based on the recently introduced Gumbel AlphaZero (GAZ), we propose our algorithm GAZ 'Play-to-Plan' (GAZ PTP), in which the agent learns to find strong trajectories by planning against possible strategies of its past self. We show the effectiveness of our approach in two well-known combinatorial optimization problems, the Traveling Salesman Problem and the Job-Shop Scheduling Problem. With only half of the simulation budget for search, GAZ PTP consistently outperforms all selected single-player variants of GAZ.

## 1 INTRODUCTION

One of the reasons for the success of AlphaZero (Silver et al., 2017) is the use of a policy and value network to guide the Monte Carlo tree search (MCTS) to decrease the search tree's width and depth. Trained on state-outcome pairs, the value network develops an 'intuition' to tell from single game positions which player might win. By normalizing values in the tree to handle changing reward scales (Schadd et al., 2008; Schrittwieser et al., 2020), AlphaZero's mechanisms can be applied to single-agent (or single-player) tasks. Although powerful, the MCTS relies on value approximations which can be hard to predict (van Hasselt et al., 2016; Pohlen et al., 2018). Furthermore, without proper normalization, it can be difficult for value function approximators to adapt to small improvements in later stages of training. In recent years there has been an increasing interest in learning sequential solutions for combinatorial optimization problems (COPs) from zero knowledge via deep reinforcement learning. Particularly strong results have been achieved with policy gradient methods by using variants of *self-critical training* (Rennie et al., 2017; Kool et al., 2018; Kwon et al., 2020) where it is avoided to learn a value function at all. By baselining the gradient estimate with the outcome of rolling out a current or historical policy, actions are reinforced by how much better (or worse) an episode is compared to the rollouts. Something similar is achieved in MCTS-based algorithms for single-player tasks by computing a scalar baseline from the agent's historical performance. The reward of the original task is reshaped to a binary $\pm 1$ outcome indicating whether an episode has exceeded this baseline or not (Laterre et al., 2018; Schmidt et al., 2019; Mandhane et al., 2022). This *self-competition* brings the original single-player task closer to a two-player game and bypasses the need for in-tree value scaling during training. The scalar baseline against which the agent is planning must be carefully chosen as it should neither be too difficult nor too easy to outperform. Additionally, in complex problems, a single scalar value holds limited information about the instance at hand and the agent's strategies for reaching the threshold performance.

In this paper, we follow the idea of self-competition in AlphaZero-style algorithms for deterministic single-player sequential planning problems. Inspired by the original powerful 'intuition' of

AlphaZero to evaluate board positions, we propose to evaluate states in the value function not by comparing them against a scalar threshold but directly against states at similar timesteps coming from a historical version of the agent. The agent learns by reasoning about potential strategies of its past self. We summarize our contributions as follows: (i) We assume that in a self-competitive framework, the scalar outcome of a trajectory $\zeta$ under some baseline policy is less informative for tree-based planning than $\zeta$'s intermediate states. Our aim is to put the flexibility of rollouts in self-critical training into a self-competitive framework while maintaining the policy's information in intermediate states of the rollout. We motivate this setup from the viewpoint of advantage baselines, show that policy improvements are preserved, and arrive at a simple instance of gamification where two players start from the same initial state, take actions in turn and aim to find a better trajectory than the opponent. (ii) We propose the algorithm GAZ Play-to-Plan (GAZ PTP) based on Gumbel AlphaZero (GAZ), the latest addition to the AlphaZero family, introduced by Danihelka et al. (2022). An agent plays the above game against a historical version of itself to improve a policy for the original single-player problem. The idea is to allow only one player in the game to employ MCTS and compare its states to the opponent's to guide the search. Policy improvements obtained through GAZ's tree search propagate from the game to the original task.

We show the superiority of GAZ PTP over single-player variants of GAZ on two COP classes, the Traveling Salesman Problem and the standard Job-Shop Scheduling Problem. We compare GAZ PTP with different single-player variants of GAZ, with and without self-competition. We consistently outperform all competitors even when granting GAZ PTP only half of the simulation budget for search. In addition, we reach competitive results for both problem classes compared to benchmarks in the literature.

## 2 RELATED WORK

**Gumbel AlphaZero** In GAZ, Danihelka et al. (2022) redesigned the action selection mechanisms of AlphaZero and MuZero (Silver et al., 2017; Schrittwieser et al., 2020). At the root node, actions to explore are sampled without replacement using the Gumbel-Top-k trick (Vieira; Kool et al., 2019b). Sequential halving (Karnin et al., 2013) is used to distribute the search simulation budget among the sampled actions. The singular action remaining from the halving procedure is selected as the action to be taken in the environment. At non-root nodes, action values of unvisited nodes are completed by a value interpolation, yielding an updated policy. An action is then selected deterministically by matching this updated policy to the visit count distribution (Grill et al., 2020). This procedure theoretically guarantees a policy improvement for correctly estimated action values, both for the root action selection and the updated policy at non-root nodes. Consequently, the principled search of GAZ works well even for a small number of simulations, as opposed to AlphaZero, which might perform poorly if not all actions are visited at the root node. Similarly to MuZero, GAZ normalizes action values with a min-max normalization based on the values found during the tree search to handle changing and unbounded reward scales. However, if for example a node value is overestimated, the probability of a good action might be reduced even when all simulations reach the end of the episode. Additionally, value function approximations might be challenging if the magnitude of rewards changes over time or must be approximated over long time horizons (van Hasselt et al., 2016; Pohlen et al., 2018).

**Self-critical training** Rennie et al. (2017) introduce self-critical training, a policy gradient method that baselines the REINFORCE (Williams, 1992) gradient estimator with the reward obtained by rolling out the current policy greedily. As a result, trajectories outperforming the greedy policy are given positive weight while inferior ones are suppressed. Self-critical training eliminates the need for learning a value function approximator (and thus all innate training challenges) and reduces the variance of the gradient estimates. The agent is further provided with an automatically controlled curriculum to keep improving. Self-critical methods have shown great success in neural combinatorial optimization. Kool et al. (2018) use a greedy rollout of a periodically updated best-so-far policy as a baseline for routing problems. Kwon et al. (2020) and Kool et al. (2019a) bundle the return of multiple sampled trajectories to baseline the estimator applied to various COPs. The idea of using rollouts to control the agent's learning curriculum is hard to transfer one-to-one to MCTS-based algorithms, as introducing a value network to avoid full Monte Carlo rollouts in the tree search was exactly one of the great strengths in AlphaZero (Silver et al., 2017).

**Self-competition**   Literature is abundant on learning by creating competitive environments for (practical) single-player problems, e.g. Bansal et al. (2017); Sukhbaatar et al. (2018); Zhong et al. (2019); Xu & Lieberherr (2020); Wang et al. (2020); Göttl et al. (2022). Reward mechanisms based on *self-competition* for AlphaZero-type algorithms are different but comparable to self-critical training in policy gradient methods. A scalar baseline is computed from the agent's historical performance against which the current agent needs to compete. A reward of $\pm 1$ is returned at the end of an episode, depending on whether the trajectory performed better than the baseline, eliminating the need for value scaling and normalization. An iterative process of improvement is created by continuously or periodically updating the baseline according to the agent's performance. As in self-critical training, the baseline provides the agent automatically with the right curriculum to improve. We use the term *self-competition* in the context of competing against historical performance to distinguish from *self-play* as in board games, where usually the latest agent version is used. Ranked Reward (Laterre et al., 2018) stores historical rewards in a buffer which is used to calculate a threshold value based on a predetermined percentile. Mandhane et al. (2022) use an exponential moving average-based scheme for the threshold and apply self-competition to maximize a constrained objective. Beloborodov et al. (2020) use a similar approach as Ranked Reward, but also allow non-binary rewards in $[-1, 1]$. The described self-competition based on combining historical performances cannot differentiate between problem instances. If an instance's difficulty is higher than usual, in extreme cases the agent might obtain a reward of $-1$ even though the found solution is optimal, thus adding significant noise to the reshaped reward (Bello et al., 2016; Laterre et al., 2018).

## 3   PRELIMINARIES

### 3.1   PROBLEM FORMULATION

We consider undiscounted Markov decision processes (MDPs) of finite problem horizon $T$ with state space $\mathcal{S}$, a finite action space $\mathcal{A}$, a reward function $r \colon \mathcal{S} \times \mathcal{A} \to \mathbb{R}$, and an initial state distribution $\rho_0$. The goal is to find a state-dependent policy $\pi$ which maximizes the expected total reward

$$G^\pi := \mathbb{E}_{s_0 \sim \rho_0} \mathbb{E}_{\zeta_0 \sim \eta^\pi(\cdot|s_0)} \left[ \sum_{t=0}^{T-1} r(s_t, a_t) \right],$$

where $\eta^\pi(\cdot|s_t)$ is the distribution over possible trajectories $\zeta_t = (s_t, a_t, \ldots, s_{T-1}, a_{T-1}, s_T)$ obtained by rolling out policy $\pi$ from state $s_t$ at timestep $t$ up to the finite horizon $T$. We use the subscript $t$ in actions and states to indicate the timestep index. For an action $a_t \sim \pi(\cdot|s_t)$ in the trajectory, the state transitions deterministically to $s_{t+1} = F(s_t, a_t)$ according to some known deterministic state transition function $F \colon \mathcal{S} \times \mathcal{A} \to \mathcal{S}$. We shortly write $a_t s_t := F(s_t, a_t)$ for a state transition in the following. With abuse of notation, we write $r(\zeta_t) := \sum_{l=0}^{T-1-t} r(s_{t+l}, a_{t+l})$ for the accumulated return of a trajectory. As usual, we denote by

$$V^\pi(s_t) := \mathbb{E}_{\zeta_t \sim \eta^\pi(\cdot|s_t)} \left[ r(\zeta_t) \right] \qquad Q^\pi(s_t, a_t) := r(s_t, a_t) + \mathbb{E}_{\zeta_{t+1} \sim \eta^\pi(\cdot|a_t s_t)} \left[ r(\zeta_{t+1}) \right]$$
$$A^\pi(s_t, a_t) := Q^\pi(s_t, a_t) - V^\pi(s_t)$$

the state-value function, action-value function and advantage function w.r.t. the policy $\pi$. Furthermore, for two policies $\pi$ and $\mu$, we define

$$V^{\pi,\mu}(s_t, s_l') := \mathbb{E}_{\substack{\zeta_t \sim \eta^\pi(\cdot|s_t) \\ \zeta_l' \sim \eta^\mu(\cdot|s_l')}} \left[ r(\zeta_t) - r(\zeta_l') \right] \tag{1}$$

as the expected difference in accumulated rewards taken over the joint distribution of $\eta^\pi(\cdot|s_t)$ and $\eta^\mu(\cdot|s_l')$. We define $Q^{\pi,\mu}(s_t, s_l'; a_t) := r(s_t, a_t) + V^{\pi,\mu}(a_t s_t, s_l')$ and $A^{\pi,\mu}(s_t, s_l'; a_t) := Q^{\pi,\mu}(s_t, s_l'; a_t) - V^{\pi,\mu}(s_t, s_l')$ analogously.

The following lemma follows directly from the definitions. It tells us that we can work with the policy $\mu$ as with any other scalar baseline, and that policy improvements are preserved. A proof is given in Appendix A.1.

**Lemma 1** *Let $\pi, \tilde{\pi}$, and $\mu$ be state-dependent policies. For any states $s_t, s_l' \in \mathcal{S}$ and action $a_t \in \mathcal{A}$, we have*

$$A^\pi(s_t, a_t) = A^{\pi,\mu}(s_t, s_l'; a_t), \text{ and} \tag{2}$$
$$\left[ \sum_{a_t} \tilde{\pi}(a_t|s_t) Q^\pi(s_t, a_t) \right] - V^\pi(s_t) = \left[ \sum_{a_t} \tilde{\pi}(a_t|s_t) Q^{\pi,\mu}(s_t, s_l'; a_t) \right] - V^{\pi,\mu}(s_t, s_l'). \tag{3}$$

### 3.2 MOTIVATION FOR THE TWO-PLAYER GAME

We now consider MDPs with episodic rewards, i.e., we assume $r(s_t, a_t) = 0$ for $t < T - 1$. The policy target and action selection within the search tree in GAZ is based on the following policy update: At some state $s$, given logits of the form $\text{logit}^\pi(a)$ for an action $a$ predicted by a policy $\pi$, an updated policy $\pi'_{\text{GAZ}}$ is obtained by setting

$$\text{logit}^{\pi'_{\text{GAZ}}}(a) = \text{logit}^\pi(a) + \sigma(\hat{A}^\pi(s, a)). \tag{4}$$

Here, $\sigma$ is some monotonically increasing linear function and $\hat{A}^\pi(s, a) := \hat{Q}^\pi(s, a) - \hat{V}^\pi(s)$ is an advantage estimation, where $\hat{V}^\pi(s)$ is a value approximation based on the output of a value network and $\hat{Q}^\pi(s, a)$ is a $Q$-value approximation coming from the tree search statistics (and is set to $\hat{V}^\pi(s)$ for unvisited actions). Note that (4) differs from the presentation in Danihelka et al. (2022) due to the additional assumption that $\sigma$ is linear, which matches its practical choice (see Appendix A.2 for a derivation). The update (4) is proven to provide a policy improvement but relies on the correctness of the value approximations $\hat{V}^\pi$.

*Our aim is to couple the ideas of self-critical training and self-competition using a historical policy. Firstly, the baseline should adapt to the difficulty of an instance via rollouts as in self-critical training. Secondly, we want to avoid losing information about intermediate states of the rollout as when condensing it to a scalar value.*

Consider some initial state $s_0 \sim \rho_0$, policy $\pi$, and a historical version $\mu$ of $\pi$. Let $\zeta_0^{\text{greedy}} = (s_0, a'_0, \ldots, s'_{T-1}, a'_{T-1}, s'_T)$ be the trajectory obtained by rolling out $\mu$ greedily. We propose to plan against $\zeta_0^{\text{greedy}}$ timestep by timestep instead of baselining the episode with $r(\zeta_0^{\text{greedy}})$, i.e., we want to approximate $V^{\pi,\mu}(s_t, s'_t)$ for any state $s_t$. By Lemma 1, we have for any policy $\pi'$

$$\sum_{a_t} \pi'(a_t|s_t)Q^\pi(s_t, a_t) \geq V^\pi(s_t) \iff \sum_{a_t} \pi'(a_t|s_t)Q^{\pi,\mu}(s_t, s'_t; a_t) \geq V^{\pi,\mu}(s_t, s'_t). \tag{5}$$

So if $\pi'$ improves policy $\pi$ baselined by $\mu$, then $\pi'$ improves $\pi$ in the original MDP. Furthermore, $\hat{A}^\pi(s_t, a_t)$ in (4) can be swapped with an approximation of $\hat{A}^{\pi,\mu}(s_t, s'_t; a_t)$ for the update in GAZ. Note that $V^{\pi,\mu}(s_t, s'_t) = \mathbb{E}_{\zeta_t, \zeta'_t}[r(\zeta_t) - r(\zeta'_t)] = \mathbb{E}_{\zeta_t, \zeta'_t}[\text{sgn}(r(\zeta_t) - r(\zeta'_t)) \cdot |r(\zeta_t) - r(\zeta'_t)|]$. Especially in later stages of training, improvements might be small and it can be hard to approximate the expected reward difference $|r(\zeta_t) - r(\zeta'_t)|$. Thus, we switch to a binary reward as in self-competitive methods and arrive at

$$V^{\pi,\mu}_{\text{sgn}}(s_t, s'_t) := \mathbb{E}_{\substack{\zeta_t \sim \eta^\pi(\cdot|s_t) \\ \zeta'_t \sim \eta^\mu(\cdot|s'_t)}} [\text{sgn}(r(\zeta_t) - r(\zeta'_t))], \tag{6}$$

the final target output of the value network in the self-competitive formulation. Note that (6) uncouples the network from the explicit requirement to predict the expected outcome of the original MDP to decide if $\pi$ is in a more advantageous state at timestep $t$ than $\mu$ (for further details, see Appendix D). We can define $Q^{\pi,\mu}_{\text{sgn}}$ analogously to (6) and obtain an improved policy $\pi'$ via

$$\text{logit}^{\pi'}(a_t) := \text{logit}^\pi(a_t) + \sigma(\hat{Q}^{\pi,\mu}_{\text{sgn}}(s_t, s'_t; a_t) - \hat{V}^{\pi,\mu}_{\text{sgn}}(s_t, s'_t)). \tag{7}$$

## 4 GAZ PLAY-TO-PLAN

### 4.1 GAME MECHANICS

Given the original MDP, the above discussion generalizes to a *two-player zero-sum perfect information game* with separated states: two players start with identical copies $s_0^1$ and $s_0^{-1}$, respectively, of an initial state $s_0 \sim \rho_0$. The superscripts 1 and $-1$ indicate the first (max-) and second (min-) player, respectively. Player 1 observes both states $(s_0^1, s_0^{-1})$, chooses an action $a_0^1$ and transitions to the next state $s_1^1 = a_0^1 s_0^1$. Afterwards, player $-1$ observes $(s_1^1, s_0^{-1})$, chooses action $a_0^{-1}$, and transitions to state $s_1^{-1} = a_0^{-1} s_0^{-1}$. Then, player 1 observes $(s_1^1, s_1^{-1})$, chooses action $a_1^1$, and in this manner both players take turns until they arrive at terminal states $s_T^1$ and $s_T^{-1}$. Given the resulting trajectory $\zeta_0^p$ for player $p \in \{1, -1\}$, the outcome $z$ of the game is set to $z = 1$ if $r(\zeta_0^1) \geq r(\zeta_0^{-1})$ (player 1 wins) and $z = -1$ otherwise (player 1 loses). In case of equality, player 1 wins to discourage player $-1$ from simply copying moves.

---

**Algorithm 1:** GAZ Play-to-Plan (GAZ PTP) Training

---

**Input:** $\rho_0$: initial state distribution; $\mathcal{J}_{\text{arena}}$: set of initial states sampled from $\rho_0$

**Input:** $0 \leq \gamma < 1$: self-play parameter

Init policy replay buffer $\mathcal{M}_\pi \leftarrow \emptyset$ and value replay buffer $\mathcal{M}_V \leftarrow \emptyset$

Init parameters $\theta, \nu$ for policy net $\pi_\theta \colon \mathcal{S} \to \Delta\mathcal{A}$ and value net $V_\nu \colon \mathcal{S} \times \mathcal{S} \to [-1, 1]$

Init 'best' parameters $\theta^B \leftarrow \theta$

**foreach** episode **do**

    Sample initial state $s_0 \sim \rho_0$ and set $s_0^p \leftarrow s_0$ for $p = 1, -1$

    Assign learning actor to player position: $l \leftarrow \text{random}(\{1, -1\})$

    Set greedy actor's policy $\mu \leftarrow \begin{cases} \pi_\theta & \text{with probability } \gamma, \\ \pi_{\theta^B} & \text{with probability } 1 - \gamma \end{cases}$

    **for** $t = 0, \ldots, T-1$ **do**

        **for** player $p = 1, -1$ **do**

            **if** player $p \neq l$ **then**

                Take greedy action $a_t^p$ according to policy $\mu(s_t^p)$ and receive new state $s_{t+1}^p$

            **else**

                Perform policy improvement $\mathcal{I}$ with MCTS using $V_\nu(\cdot, \cdot)$ and $\pi_\theta(\cdot)$ for player $p$

                    where in tree, player $-p$ samples (resp. chooses greedily) actions from $\mu$

                Receive improved policy $\mathcal{I}\pi(s_t^p)$, action $a_t^p$ and new state $s_{t+1}^p$

                Store $(s_t^p, \mathcal{I}\pi(s_t^p))$ in $\mathcal{M}_\pi$

    Have trajectories $\zeta^p \leftarrow (s_0^p, a_0^p, \ldots, s_{T-1}^p, a_{T-1}^p, s_T^p)$ for players $p \in \{1, -1\}$

    $z \leftarrow \begin{cases} 1 & \text{if } r(\zeta^1) \geq r(\zeta^{-1}), \\ -1 & \text{else} \end{cases}$     ▷ game outcome from perspective of player 1

    Store tuples $(s_t^1, s_t^{-1}, z)$ and $(s_t^{-1}, s_{t+1}^1, -z)$ in $\mathcal{M}_V$ for all timesteps $t$

    Periodically update $\theta^B \leftarrow \theta$ if $\sum_{s_0 \in \mathcal{J}_{\text{arena}}} \left( r(\zeta_{0, \pi_\theta}^{\text{greedy}}) - r(\zeta_{0, \pi_{\theta^B}}^{\text{greedy}}) \right) > 0$

---

## 4.2 Algorithm

Algorithm 1 summarizes our approach for improving the agent's performance, and we provide an illustration in the appendix in Figure 2. In each episode, a 'learning actor' improves its policy by playing against a 'greedy actor', which is a historical greedy version of itself. We elaborate on the main parts in the following (more details can be found in Appendix B).

**Network and training** An agent maintains a state-dependent policy network $\pi_\theta \colon \mathcal{S} \to \Delta\mathcal{A}$ and value network $V_\nu \colon \mathcal{S} \times \mathcal{S} \to [-1, 1]$ parameterized by $\theta$ and $\nu$, respectively, where $\Delta\mathcal{A}$ is the probability simplex over actions and $V_\nu$ serves as a function approximator of (6). The agent generates experience in each episode for training the networks by playing the game in Section 4.1 against a frozen historically best policy version $\pi_{\theta^B}$ of itself. Initially, $\theta$ and $\theta^B$ are equal. The networks are trained from experience replay as in AlphaZero, where we store value targets from the perspective of both players and only policy targets coming from the learning actor.

**Choice of players** The learning actor and greedy actor are randomly assigned to player positions at the beginning of each episode. The greedy actor simply chooses actions greedily according to its policy $\pi_{\theta^B}$. In contrast, the learning actor chooses actions using the search and action selection mechanism of GAZ to improve its policy and dominate over $\pi_{\theta^B}$. We found the policy to become stronger when the learning actor makes its moves sometimes after and sometimes before the greedy actor. At test time, we fix the learning actor to the position of player 1.

**GAZ tree search for learning actor** The learning actor uses MCTS to improve its policy and dominate over the greedy behavior of $\pi_{\theta^B}$. We use the AlphaZero variant (as opposed to MuZero) in Danihelka et al. (2022) for the search, i.e., we do not learn models for the reward and transition dynamics. The search tree is built through several simulations, as usual, each consisting of the phases

*selection*, *expansion*, and *backpropagation*. Each node in the tree is of the form $\mathcal{N} = (s^1, s^{-1}, k)$, consisting of both players' states, where $k \in \{1, -1\}$ indicates which player's turn it is. Each edge $(\mathcal{N}, a)$ is a node paired with a certain action $a$. We apply the tree search of GAZ similarly to a two-player board game, with two major modifications:

**(i) Choice of actions:** For the sake of explanation, let us assume that $l = 1$, i.e., the learning actor takes the position of player 1. In the *selection* phase, an action is chosen at any node as follows: If the learning actor is to move, we regularly choose an action according to GAZ's tree search rules, which are based on the completed $Q$-values. If it is the greedy actor's turn, there are two ways to use the policy $\pi_{\theta^B}$: we either always *sample*, or always *choose greedily* an action from $\pi_{\theta^B}$. By sampling a move from $\pi_{\theta^B}$, the learning actor is forced to plan not against only one (possibly suboptimal) greedy trajectory but also against other potential trajectories. This encourages exploration (see Figure 1 in the experimental results), but a single action of the learning actor can lead to separate branches in the search tree. On the other hand, greedily choosing an action is computationally more efficient, as only the states in the trajectory $\zeta_{0,\pi_{\theta^B}}^{\text{greedy}}$ are needed for comparison in the MCTS (see Appendix B.6 for notes on efficient implementation). However, if the policy $\pi_{\theta^B}$ is too strong or weak, the learning actor might be slow to improve in the beginning.

We refer to the variant of sampling actions for the greedy actor in the tree search as GAZ PTP 'sampled tree' (GAZ PTP ST) and as GAZ PTP 'greedy tree' (GAZ PTP GT) in case actions are chosen greedily.

**(ii) Backpropagation:** Actions are chosen as above until a leaf edge $(\mathcal{N}, a)$ with $\mathcal{N} = (s^1, s^{-1}, k)$ is reached. When a leaf edge $(\mathcal{N}, a)$ is *expanded* and $k = -l = -1$, the learning actor is to move in the new node $\tilde{\mathcal{N}} = (s^1, as^{-1}, 1)$. We proceed by querying the policy and value network to obtain $\pi_\theta(s^1)$ and $v := V_\nu(s^1, as^{-1})$. The new node $\tilde{\mathcal{N}}$ is added to the tree and the value approximation $v$ is *backpropagated* up the search path. If $k = 1$, i.e. the turn of the greedy actor in $\tilde{\mathcal{N}} = (as^1, s^{-1}, -1)$, we only query the policy network $\pi_{\theta^B}(s^{-1})$, choose an action $\tilde{a}$ from it (sampled or greedy), and *directly* expand the edge $(\tilde{N}, \tilde{a})$. In particular, we do not backpropagate a value approximation of state pairs where the greedy actor is to move. Nodes, where it's the greedy actor's turn, are similar to *afterstates* (Sutton & Barto, 2018), which represent chance transitions in stochastic environments. We further illustrate the procedure in Appendix B.2.

**Arena mode** To ensure we always play against the strongest greedy actor, we fix at the beginning of the training a large enough 'arena' set of initial states $\mathcal{J}_{\text{arena}} \sim \rho_0$ on which periodically $\pi_\theta$ and $\pi_{\theta^B}$ are pitted against each other. For each initial state $s_0 \in \mathcal{J}_{\text{arena}}$, the policies $\pi_\theta$ and $\pi_{\theta^B}$ are unrolled greedily to obtain the trajectories $\zeta_{0,\pi_\theta}^{\text{greedy}}$ and $\zeta_{0,\pi_{\theta^B}}^{\text{greedy}}$, respectively. We replace $\theta^B$ with $\theta$ if

$$\sum_{s_0 \in \mathcal{J}_{\text{arena}}} \left( r(\zeta_{0,\pi_\theta}^{\text{greedy}}) - r(\zeta_{0,\pi_{\theta^B}}^{\text{greedy}}) \right) > 0.$$

**Self-play parameter** The parameters $\theta, \theta^B$ might be initialized disadvantageously (worse than uniformly random play), so that the learning actor generates no useful experience while maintaining a greedy policy worse than $\pi_{\theta^B}$. We especially experienced this behavior when using a small number of simulations for the tree search. To mitigate this and to stabilize training, we switch to *self-play* in an episode with a small nonzero probability $\gamma$, i.e., the greedy actor uses the non-stationary policy $\pi_\theta$ instead of $\pi_{\theta^B}$.

## 5 EXPERIMENTS

We evaluate our algorithm on the Traveling Salesman Problem (TSP) and the standard Job-Shop Scheduling Problem (JSSP). For this purpose, we train the agent on randomly generated instances to learn heuristics and generalize beyond presented instances.

**Experimental goal** There is a plethora of deep reinforcement learning approaches for training heuristic solvers for COPs, including Bello et al. (2016); Deudon et al. (2018); Kool et al. (2018); Xing et al. (2020); da Costa et al. (2020); Kool et al. (2022) for the TSP and Zhang et al. (2020); Park et al. (2021a;b); Tassel et al. (2021) for the JSSP. Most of them are problem-specific and especially concentrate on how to optimize (graph) neural network-based architectures, state representations,

and reward shaping. Additionally, learned solvers often focus on short evaluation times while accepting a high number of training instances. Our main goal is to show the performance of GAZ PTP compared with applying GAZ in a single-player way. Nevertheless, we also present results of recent learned solvers to put our approach into perspective. An in-depth comparison is difficult for several reasons. (i) Running MCTS is slower than a single (possibly batch-wise) rollout of a policy network. As we do not aim to present an optimized learned solver for TSP or JSSP, we omit comparisons of inference times. (ii) The network architecture is kept simple to be adjustable for both problem classes. (iii) We give only episodic rewards, even if there would be (canonical) intermediate rewards. Furthermore, we do not include heuristics in state representations. (iv) Eventually, the agent starts from zero knowledge and runs for a comparably lower number of episodes: 100k for TSP (for comparison, Kool et al. (2018) trains on 128M trajectories), 20k for JSSP (40k trajectories in Zhang et al. (2020)).

## 5.1 SINGLE-PLAYER VARIANTS

We compare our approach GAZ PTP with the following single-player GAZ variants:

**GAZ Single Vanilla** We apply GAZ for single-agent domains to the original MDP. For the value target, we predict the final outcome of the episode, i.e., we train the value network $V_\nu \colon \mathcal{S} \to \mathbb{R}$ on tuples of the form $(s_t, r(\zeta_0))$ for a trajectory $\zeta_0 = (s_0, a_0, \ldots, a_{T-1}, s_T)$. We transform the $Q$-values with a min-max normalization based on the values found inside the tree search to cope with different reward scales in MCTS (Schrittwieser et al., 2020; Danihelka et al., 2022). For details, see Appendix C.1.3.

**GAZ Single N-Step** Same as 'GAZ Single Vanilla', except: we bootstrap $N$ steps into the future for the value target, as used in MuZero (Schrittwieser et al., 2020) and its variants (Hubert et al., 2021; Antonoglou et al., 2022). As we do not assume intermediate rewards, this is equivalent to predicting the undiscounted root value of the search tree $N$ steps into the future. We set $N = 20$ in all instances.

**GAZ Greedy Scalar** This is a self-competitive version with a scalar baseline, which can differentiate between instances as in a self-critical setting. At the start of each episode, a full greedy rollout $\zeta_0^{\text{greedy}}$ from the initial state $s_0$ is performed with a historical policy to obtain an instance-specific baseline $R := r(\zeta_0^{\text{greedy}}) \in \mathbb{R}$. At test time, $\max\{R, \text{learning actor's outcome}\}$ is taken as the result in the original MDP. A listing of the algorithm is provided in Appendix C.1.4.

## 5.2 GENERAL SETUP

We outline the common experimental setup for both TSP and JSSP. Full details and specifics can be found in Appendix C.1.

**GAZ MCTS** We closely follow the original work of Danihelka et al. (2022) for the choice of the function $\sigma$ in (4) and the completion of $Q$-values. The number of simulations allowed from the search tree's root node is critical since a higher simulation budget generally yields more accurate value estimates. Although two edges are expanded in a single simulation in the modified tree search of GAZ PTP, the learning actor considers only one additional action in the expansion step because the greedy actor's moves are considered as environment transitions. Nevertheless, we grant the single-player variants twice as many simulations at the root node as GAZ PTP in all experiments.

**Network architecture** The policy and value network share a common encoding part $f \colon \mathcal{S} \to \mathbb{R}^d$ mapping states to some latent space $\mathbb{R}^d$. A policy head $g \colon \mathbb{R}^d \to \Delta \mathcal{A}$ and value head $h \colon \mathbb{R}^d \times \mathbb{R}^d \to [-1, 1]$ are stacked on top of $f$ such that $\pi_\theta(s) = g(f(s))$ and $V_\nu(s, s') = h(f(s), f(s'))$. For both problems, $f$ is based on the Transformer architecture (Vaswani et al., 2017) and the policy head $g$ uses a pointing mechanism as in (Bello et al., 2016) and (Kool et al., 2018). For GAZ PTP and all single-player variants, the architecture of $f$ is identical, and $g, h$ similar (see Appendix C.1.5).

Table 1: Results for TSP and JSSP. 'Num Sims' refers to the number of simulations starting from the root of the search tree.

| | Method | Num Sims | Obj. | Gap | Obj. | Gap | Obj. | Gap |
|---|---|---|---|---|---|---|---|---|
| | | | $n = 20$ | | $n = 50$ | | $n = 100$ | |
| **TSP** | Optimal (Concorde) | — | 3.84 | 0.00% | 5.70 | 0.00% | 7.76 | 0.00% |
| | Kool et al. (2018) (gr.) | — | 3.85 | 0.34% | 5.80 | 1.76% | 8.12 | 4.53% |
| | GAZ Single Vanilla | 200 | 3.86 | 0.58% | 6.06 | 6.36% | 9.87 | 27.15% |
| | GAZ Single N-Step | 200 | 3.86 | 0.58% | 5.92 | 3.98% | 9.14 | 17.72% |
| | GAZ Greedy Scalar | 200 | 3.87 | 0.83% | 6.02 | 5.74% | 11.21 | 44.43% |
| | GAZ PTP ST | 100 | 3.84 | 0.19% | 5.81 | 1.90% | 8.16 | 5.11% |
| | GAZ PTP GT | 100 | 3.84 | 0.17% | 5.78 | 1.55 % | 8.01 | 3.16% |
| | GAZ PTP ST | greedy[1] | 3.86 | 0.74% | 5.90 | 3.63% | 8.35 | 7.60% |
| | GAZ PTP GT | greedy[1] | 3.86 | 0.61% | 5.82 | 2.15% | 8.10 | 4.32% |
| | | | $15 \times 15$ | | $20 \times 20$ | | $30 \times 20$ | |
| **JSSP** | Upper Bound | — | 1228.9 | 0.0% | 1617.3 | 0.0% | 1921.3 | 0.0% |
| | Zhang et al. (2020) | — | 1547.4 | 26.0% | 2128.1 | 31.6% | 2603.9 | 33.6% |
| | GAZ Single Vanilla | 100 | 1585.8 | 29.0% | 2062.4 | 27.5% | 2579.1 | 34.4% |
| | GAZ Single N-Step | 100 | 2038.4 | 66.0% | 3604.6 | 123.0% | 4111.2 | 114.6% |
| | GAZ Greedy Scalar | 100 | 1447.9 | 17.8% | 4721.5 | 191.9% | 6184.3 | 222.1% |
| | GAZ PTP ST | 50 | 1432.0 | 16.6% | 1973.2 | 22.0% | 2539.1 | 32.3% |
| | GAZ PTP GT | 50 | 1455.4 | 18.4% | 1961.4 | 21.3% | 2506.7 | 30.6% |
| | GAZ PTP ST | greedy[1] | 1505.7 | 22.6% | 1993.0 | 23.2% | 2601.8 | 35.6% |
| | GAZ PTP GT | greedy[1] | 1478.8 | 20.3% | 2003.7 | 23.9% | 2584.2 | 34.6% |

## 5.3 RESULTS

Results for TSP and JSSP are summarized in Table 1. For TSP, we also list the greedy results of the self-critical method of Kool et al. (2018). They train a similar attention model autoregressively with a greedy rollout baseline. For JSSP, we include the results of the L2D method of Zhang et al. (2020), who propose a single-agent approach taking advantage of the disjunctive graph representation of JSSP. There exist optimized approaches for TSP and JSSP, which achieve even better results. Since it is not our aim to present a new state-of-the-art solver, an exhaustive comparison is not necessary. All experiments, except for the robustness experiments below, are seeded with 42.

**Traveling Salesman Problem** The TSP is a fundamental routing problem that, given a graph, asks for a node permutation (a complete *tour*) with minimal edge weight. We focus on the two-dimensional Euclidean case in the unit square $[0, 1]^2 \subseteq \mathbb{R}^2$, and train on small- to medium-sized instances with $n = 20$, 50 and 100 nodes. Tours are constructed sequentially by choosing one node to visit after the other. We run the agent for 100k episodes (see Appendix C.2 for further details). We report the average performance of all GAZ variants on the 10,000 test instances of Kool et al. (2018). Optimality gaps are calculated using optimal solutions found by the Concorde TSP solver (Applegate et al., 2006). Both GAZ PTP ST and GAZ PTP GT consistently outperform all single-player variants, and obtain strong results even when simply unrolling the learned policy greedily (see 'Num Sims = greedy' in Table 1). GAZ PTP GT yields better results than GAZ PTP ST across all instances, especially when rolling out the learned policy greedily. While all methods perform well on $n = 20$, the single-player variants strongly underperform on $n = 50$ and $n = 100$. GAZ Greedy Scalar fails to improve early in the training process for $n = 100$.

**Job-Shop Scheduling Problem** The JSSP is an optimization problem in operations research, where we are given $k$ jobs consisting of individual operations, which need to be scheduled on $m$ machines. We focus on the standard case, where there is a bijection between the machines and the operations of a job. The objective is to find a *schedule* with a minimum *makespan*, i.e., the time

---

[1]We greedily unroll the policy which was trained with 100 (resp. 50 for JSSP) simulations.

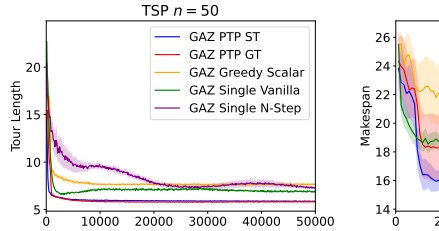 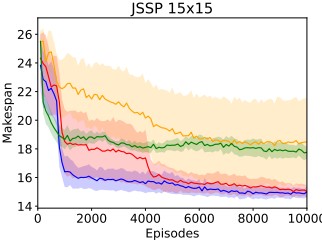 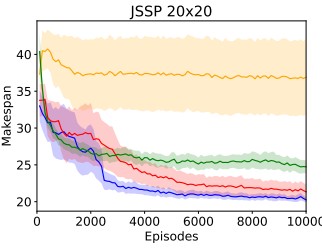

Figure 1: Mean training performance on TSP and JSSP, run with 4 distinct seeds $\in \{42, 43, 44, 45\}$. Shades denote standard errors. We omit the results for GAZ Single N-Step on JSSP for readability.

when all jobs are finished (see Appendix C.3 for details). The size of a problem instance is denoted by $k \times m$ (jobs $\times$ machines). To construct a schedule, we iteratively choose unfinished jobs of which to process the next operation. The agent is run for 20k episodes. We report results on instances of medium size $15 \times 15$, $20 \times 20$, and $30 \times 20$ of the well-known Taillard benchmark set (Taillard, 1993), consisting of 10 instances for each problem size. Optimality gaps are calculated with respect to the best upper bounds found in the literature (see Appendix C.3.3). As for TSP, our method outperforms all single-player variants of GAZ. In contrast to TSP, the performance of GAZ PTP ST and GAZ PTP GT is comparable. Due to the reduced network evaluations, GAZ PTP GT is generally more favorable. GAZ Greedy Scalar yields strong results only for $15 \times 15$ but fails to learn on larger instances, similar to GAZ Single N-Step.

**Reproducibility**  We further evaluate the robustness and plausibility on four different seeds for TSP $n = 50$ and JSSP $15 \times 15$ and $20 \times 20$ in Figure 1. To lower the computational cost, we reduce the number of episodes on TSP to 50k and JSSP to 10k, with a simulation budget of 100 for TSP (70 for JSSP) for single-player variants, and 50 for TSP (35 for JSSP) for GAZ PTP. The different seeding leads to only small variations in performance for TSP. JSSP is more challenging. Again, we can observe that GAZ PTP outperforms the single-player variants across all seeds. Especially, GAZ PTP escapes bad initializations early in the training process and becomes stable. The in-tree sampling of GAZ PTP ST encourages exploration, leading to swift early improvements.

**Value estimates as baselines**  By comparing pairs of states to guide the tree search, we provide the network some freedom to model the problem without relying explicitly on predicted expected objectives. In Appendix D, we compare our approach with using *value estimates* of a historical policy as baselines in the advantage function of GAZ's tree search.

**Limitations**  In contrast to GAZ PTP GT, the search space increases exponentially for the variant GAZ PTP ST, and an additional network evaluation is needed for $\pi_{\theta^B}(\cdot)$ in each MCTS simulation (see Appendix B.6). Nevertheless, our empirical results show that both GAZ PTP variants perform well in general with a small number of simulations. In this paper, we only consider problem classes with constant episode length. However, the methodology presented extends to varying episode lengths: Once a player finishes, only its terminal state is considered in the remaining tree search (i.e., within the two-player game, the unfinished player keeps its turn). Furthermore, even though GAZ's policy improvements propagate from the two-player setup to the original single-player problem by Lemma 1, an improved policy does not necessarily imply improved greedy behavior. Especially in later stages of training, it can take thousands of steps until the distribution of $\pi_\theta$ is sharp enough for updating the parameters $\theta^B$.

## 6 CONCLUSION

We introduced GAZ PTP, a self-competitive method combining greedy rollouts as in self-critical training with the planning dynamics of two-player games. The self-critical transformation of a deterministic single-player task does not alter the theoretical policy improvement guarantees obtained through the principled search of GAZ. Experiments on the TSP and JSSP confirm that our method learns strong policies in the original task with a low number of search simulations.

## 7 REPRODUCIBILITY STATEMENT

We provide information about network architectures, hyperparameters and training details in the appendix. Our code in PyTorch (Paszke et al., 2017) is available on `https://github.com/grimmlab/policy-based-self-competition`.

### ACKNOWLEDGMENTS

This work was funded by the Deutsche Forschungsgemeinschaft (DFG, German Research Foundation) – 466387255 – within the Priority Programme "SPP 2331: Machine Learning in Chemical Engineering". The authors gratefully acknowledge the Leibniz Supercomputing Centre for providing computing time on its Linux-Cluster.

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

## A  PROOFS

### A.1  PROOF OF LEMMA 1

*Lemma:* Let $\pi, \tilde{\pi}$ and $\mu$ be state-dependent policies. For any states $s_t, s'_l \in \mathcal{S}$ and action $a_t \in \mathcal{A}$, we have

$$A^\pi(s_t, a_t) = A^{\pi,\mu}(s_t, s'_l; a_t), \text{ and}$$
$$\left[ \sum_{a_t} \tilde{\pi}(a_t|s_t) Q^\pi(s_t, a_t) \right] - V^\pi(s_t) = \left[ \sum_{a_t} \tilde{\pi}(a_t|s_t) Q^{\pi,\mu}(s_t, s'_l; a_t) \right] - V^{\pi,\mu}(s_t, s'_l).$$

*Proof:* We recall the definitions for the sake of clarity:

$$V^{\pi,\mu}(s_t, s'_l) := \mathbb{E}_{\substack{\zeta_t \sim \eta^\pi(\cdot|s_t) \\ \zeta'_l \sim \eta^\mu(\cdot|s'_l)}} [r(\zeta_t) - r(\zeta'_l)] \qquad Q^{\pi,\mu}(s_t, s'_l; a_t) := r(s_t, a_t) + V^{\pi,\mu}(a_t s_t, s'_l)$$
$$A^{\pi,\mu}(s_t, s'_l; a_t) := Q^{\pi,\mu}(s_t, s'_l; a_t) - V^{\pi,\mu}(s_t, s'_l)$$

Note that

$$V^\pi(s_t) = \mathbb{E}_{\zeta_t \sim \eta^\pi(\cdot|s_t)} [r(\zeta_t)] = \mathbb{E}_{\zeta'_l \sim \eta^\mu(\cdot|s'_l)} \mathbb{E}_{\zeta_t \sim \eta^\pi(\cdot|s_t)} [r(\zeta_t)] \tag{8}$$
$$= \mathbb{E}_{\substack{\zeta_t \sim \eta^\pi(\cdot|s_t) \\ \zeta'_l \sim \eta^\mu(\cdot|s'_l)}} [r(\zeta_t)], \tag{9}$$

by the law of iterated expectations and because the realization of a trajectory following $\pi$ does not depend on $\mu$ and vice versa. Hence by linearity of expectations

$$V^\pi(s_t) - V^\mu(s'_l) = \mathbb{E}_{\substack{\zeta_t \sim \eta^\pi(\cdot|s_t) \\ \zeta'_l \sim \eta^\mu(\cdot|s'_l)}} [r(\zeta_t) - r(\zeta'_l)] = V^{\pi,\mu}(s_t, s'_l), \tag{10}$$

and it follows that

$$A^\pi(s_t, a_t) = Q^\pi(s_t, a_t) - V^\pi(s_t) \tag{11}$$
$$= Q^\pi(s_t, a_t) - V^\mu(s'_l) - (V^\pi(s_t) - V^\mu(s'_l)) \tag{12}$$
$$= r(s_t, a_t) + V^\pi(a_t s_t) - V^\mu(s'_l) - (V^\pi(s_t) - V^\mu(s'_l)) \tag{13}$$
$$\overset{(10)}{=} r(s_t, a_t) + V^{\pi,\mu}(a_t s_t, s'_l) - V^{\pi,\mu}(s_t, s'_l) \tag{14}$$
$$= Q^{\pi,\mu}(s_t, s'_l; a_t) - V^{\pi,\mu}(s_t, s'_l) \tag{15}$$
$$= A^{\pi,\mu}(s_t, s'_l; a_t). \tag{16}$$

Furthermore, we have

$$\left[\sum_{a_t}\tilde{\pi}(a_t|s_t)Q^\pi(s_t,a_t)\right] - V^\pi(s_t) = \left[\sum_{a_t}\tilde{\pi}(a_t|s_t)\big(r(s_t,a_t)+V^\pi(a_ts_t)\big)\right] - V^\pi(s_t) \quad (17)$$

$$= \left[\sum_{a_t}\tilde{\pi}(a_t|s_t)\big(r(s_t,a_t)+V^\pi(a_ts_t)-V^\mu(s_l') \quad (18)\right.$$

$$\left. + V^\mu(s_l')\big)\right] - V^\pi(s_t)$$

$$\stackrel{(10)}{=} \left[\sum_{a_t}\tilde{\pi}(a_t|s_t)\big(\underbrace{r(s_t,a_t)+V^{\pi,\mu}(a_ts_t,s_l')}_{=Q^{\pi,\mu}(s_t,s_l';a_t)} \quad (19)\right.$$

$$\left. + V^\mu(s_l')\big)\right] - V^\pi(s_t)$$

$$= \left[\sum_{a_t}\tilde{\pi}(a_t|s_t)Q^{\pi,\mu}(s_t,s_l';a_t)\right] \quad (20)$$

$$+ \underbrace{\left[\sum_{a_t}\tilde{\pi}(a_t|s_t)V^\mu(s_l')\right]}_{=V^\mu(s_l')} - V^\pi(s_t)$$

$$= \left[\sum_{a_t}\tilde{\pi}(a_t|s_t)Q^{\pi,\mu}(s_t,s_l';a_t)\right] - (V^\pi(s_t)-V^\mu(s_l'))$$
$$(21)$$

$$= \left[\sum_{a_t}\tilde{\pi}(a_t|s_t)Q^{\pi,\mu}(s_t,s_l';a_t)\right] - V^{\pi,\mu}(s_t,s_l'). \quad (22)$$

$$\square$$

## A.2 DERIVATION OF THE LOGIT UPDATE (4)

In Danihelka et al. (2022), the improved policy $\pi'_{\text{GAZ}}$ is obtained by the logit update

$$\text{logit}^{\pi'_{\text{GAZ}}}(a) := \text{logit}^\pi(a) + \sigma(\hat{Q}^\pi(s,a)). \quad (23)$$

Subtracting the constant $\sigma(\hat{V}^\pi(s))$ does not alter the subsequent softmax-output of the logits, so we can equivalently set

$$\text{logit}^{\pi'_{\text{GAZ}}}(a) = \text{logit}^\pi(a) + \sigma(\hat{Q}^\pi(s,a)) - \sigma(\hat{V}^\pi(s)) \stackrel{\sigma \text{ linear}}{=} \text{logit}^\pi(a) + \sigma(\hat{Q}^\pi(s,a)-\hat{V}^\pi(s))$$
$$(24)$$

$$= \text{logit}^\pi(a) + \sigma(\hat{A}^\pi(s,a)). \quad (25)$$

## B ADDITIONAL ALGORITHM DETAILS

### B.1 A NOTE ON SELF-PLAY IN THE TWO-PLAYER GAME

The mechanics of the proposed two-player game differ from classical board games such as chess or Go, as a player's action does not influence the opponent's state. We can interpret trajectories in the original MDP as strategies in the game: if some trajectory $\alpha$ gives a higher final reward than trajectory $\beta$, its corresponding strategy $\alpha$ *weakly dominates* strategy $\beta$ (Leyton-Brown & Shoham, 2008). This is desired and means that playing $\alpha$ will always yield an outcome $z$ at least as good as playing $\beta$, no matter what the opponent does. Usually, in AlphaZero-type self-play for board games, both players choose their moves utilizing MCTS. By limiting the tree search to the learning actor, we can distill even incremental policy improvements to the policy network effectively.

### B.2 MODIFIED TREE SEARCH

We provide a schematic view of the proposed game in Figure 2, and an illustration of the modified *selection, expansion* and *backpropagation* in Figure 3. As outlined in Section 4.2, we do not

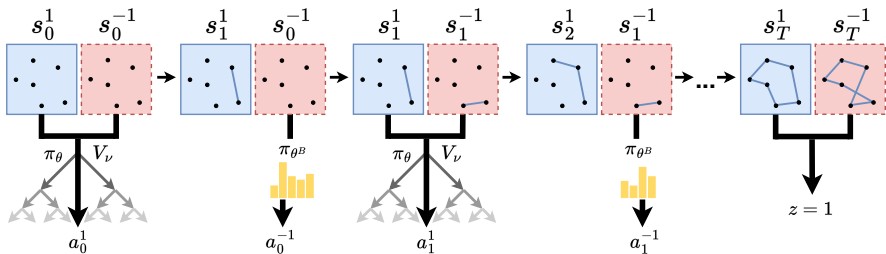

Figure 2: Schematic view of the proposed game in an example instance of the Traveling Salesman Problem: The learning actor is player $1$ (solid outline, blue), the greedy actor is player $-1$ (dashed outline, red). The learning actor chooses moves via MCTS, taking into account the states of both players, whereas the greedy actor moves greedily with respect to only its own state.

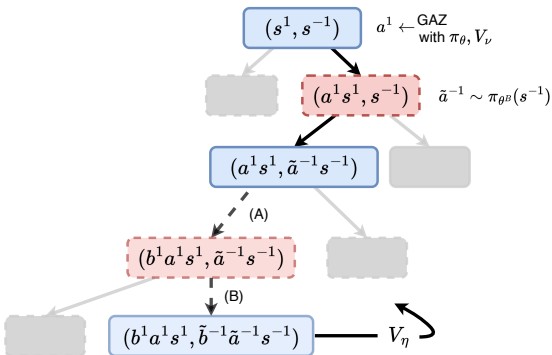

Figure 3: Example MCTS for the learning actor, when sampling actions for the greedy actor: In nodes with solid lines (blue), the learning actor (player 1) is to move, and in nodes with dashed lines (red), the greedy actor (player -1) is to move. In the selection phase (solid arrows), an action is selected in solid nodes according to the search principles of GAZ, based on $\pi_\theta$ and the completed $Q$-values. In nodes with dashed lines, an action is sampled from $\pi_{\theta^B}$. Suppose the dashed edge (A) is expanded in the expansion phase, then it is the greedy actor's turn in the following node: an action is sampled from $\pi_{\theta^B}(\tilde{a}^{-1}s^{-1})$ and the corresponding edge (B) is immediately expanded as well. Only then is the predicted value of the following solid node backpropagated through the search path.

evaluate state pairs with $V_\nu$ in nodes where it's the greedy actor's turn. This is due to computational efficiency. Suppose the greedy actor is player $-1$, and it is the greedy actor's turn in a node $\tilde{\mathcal{N}} = (as^1, s^{-1}, -1)$, which was added to the tree in an expansion step. By the modified tree search, an action is sampled from $\pi_{\theta^B}(s^{-1})$. As the policy only depends on the player's state, $\pi_{\theta^B}(s^{-1})$ can be reused for the action sampling in a node $(\tilde{s}^1, s^{-1}, -1)$ for *any* state $\tilde{s}^1$ without additional network evaluations, as the value function does not need to be queried. This can reduce the number of network evaluations in subsequent simulations.

This is different from the experiments in Stochastic MuZero (Antonoglou et al., 2022), where the authors report significantly improved results for AlphaZero on Backgammon and 2048 when a $Q$-value is learned for afterstates (chance nodes). For GAZ PTP, we did not experience notable improvements and trade the learned $Q$-value for increased computational efficiency.

For evaluation at test time, the greedy actor's moves are not sampled in the tree search but chosen greedily to increase the search tree's depth.

### B.3 A NOTE ON THE ARENA MODE

As only the sign of the reward difference $r(\zeta_{0,\pi_\theta}) - r(\zeta_{0,\pi_{\theta^B}}^{\text{greedy}})$ is considered to compute the outcome of the game, and not its magnitude, training a non-stationary policy through self-play can exhibit

behavior where the overall performance of a policy in the original MDP gradually degrades, even though the agent gets better at beating itself. In the optimal case, we would like to replace the frozen parameters $\theta^B$ with $\theta$, if

$$\mathbb{E}_{s_0 \sim \rho_0} \mathbb{E}_{\zeta_0 \sim \eta^{\pi_\theta}(\cdot | s_0)}[r(\zeta_0)] > \mathbb{E}_{s_0 \sim \rho_0} \mathbb{E}_{\zeta'_0 \sim \eta^{\pi_{\theta^B}}(\cdot | s_0)}[r(\zeta'_0)].$$

This cannot be guaranteed alone by the binary reward in our self-competitive framework. We settle for a *fixed* arena set $\mathcal{J}_{\text{arena}}$ on which the policies are pitted *greedily*, to ensure an improved greedy actor and avoid cycling performance due to stochasticity as far as possible.

### B.4 REPLAY BUFFER

Given trajectory $\zeta_0^p = (s_0^p, a_0^p, \ldots, s_{T-1}^p, a_{T-1}^p, s_T^p)$ for player $p \in \{1, -1\}$, training data is stored in replay buffers at the end of an episode as follows: For the value network, the final game outcome $z$ is bootstrapped from the perspective of both players, i.e. we store tuples $(s_t^1, s_t^{-1}, z)$ and $(s_t^{-1}, s_{t+1}^1, -z)$. For the policy network, we store tuples $(s_t^l, \mathcal{I}\pi(s_t^l))$ only for the learning actor $l \in \{1, -1\}$, where $\mathcal{I}\pi(s_t^l)$ is the improved policy obtained through the GAZ tree search at state $s_t^l$.

### B.5 LOSS FUNCTIONS

We train the value network by minimizing the squared error $(V_\nu(s, s') - \tilde{z})^2$ for a sampled tuple $(s, s', \tilde{z})$ in the replay buffer, and the policy network by minimizing the Kullback-Leibler divergence $\text{KL}(\mathcal{I}\pi(s) \parallel \pi_\theta(s))$ for a tuple $(s, \mathcal{I}\pi(s))$.

### B.6 EFFICIENT IMPLEMENTATION OF GAZ PTP GT

Algorithm 1 is a general formulation of our method encompassing both variants in which the greedy actor samples actions in the search tree in one (GAZ PTP ST) and chooses actions greedily in the other (GAZ PTP GT). For GAZ PTP GT, only states of the greedy actor encountered in the trajectory $\zeta_{0,\pi_{\theta^B}}^{\text{greedy}} = (s_0, a'_0, \ldots, s'_{T-1}, a'_{T-1}, s'_T)$, are needed in the MCTS for the learning actor. Furthermore, the policy and value network share in practice an encoding part $f \colon \mathcal{S} \to \mathbb{R}^d$ such that $V_\nu(s, s') = h(f(s), f(s'))$ for a value head $h \colon \mathbb{R}^d \times \mathbb{R}^d \to [-1, 1]$ (see Section 5.2). Hence, at the beginning of an episode, $\zeta_{0,\pi_{\theta^B}}^{\text{greedy}}$ can be obtained once in advance of the tree search, as the policies $\pi_\theta$ and $\pi_{\theta^B}$ are independent. The states $s_0, \ldots, s'_{T-1}$ can be batched, and their latent representations $f(s_0), \ldots, f(s'_{T-1}) \in \mathbb{R}^d$ can be stored in memory, effectively reducing the number of network evaluations in each search simulation from two to one (as desired in network-guided MCTS).

## C EXPERIMENTAL DETAILS

### C.1 GENERAL SETUP

#### C.1.1 TRAINING LOOP

All algorithmic variants fit into the asynchronous training loop commonly used for MuZero, where a *learning process* receives generated trajectories, stores them in a replay buffer, and performs training steps for the network. Multiple *playing processes* generate trajectories from initial states randomly sampled on the fly, using periodically updated network checkpoints from the learning process for MCTS. We generate experience with 100 playing processes and update the network checkpoint every 100 training steps in all variants.

#### C.1.2 GAZ MCTS

Given a node $\mathcal{N}$ in the search tree, denote by $N(a)$ the visit count of the edge $(\mathcal{N}, a)$ for action $a$. We follow Danihelka et al. (2022) for the choice of the monotonically increasing linear function $\sigma$ in (4) and set

$$\sigma(q) = (c_{\text{visit}} + \max_b N(b)) \cdot c_{\text{scale}} \cdot q.$$

---

**Algorithm 2:** GAZ Greedy Scalar Training

---

**Input:** $\rho_0$: initial state distribution
**Input:** $\mathcal{J}_{\text{arena}}$: set of initial states sampled from $\rho_0$
Init policy replay buffer $\mathcal{M}_\pi \leftarrow \emptyset$ and value replay buffer $\mathcal{M}_V \leftarrow \emptyset$
Init parameters $\theta, \nu$ for policy net $\pi_\theta \colon \mathcal{S} \to \Delta\mathcal{A}$ and value net $V_\nu \colon \mathcal{S} \times \mathbb{R} \to [-1, 1]$
Init 'best' parameters $\theta^B \leftarrow \theta$
**foreach** episode **do**
    Sample initial state $s_0 \sim \rho_0$
    Perform greedy rollout using $\pi_{\theta^B}$ and obtain $R \leftarrow r(\zeta_{0,\pi_{\theta^B}}^{\text{greedy}})$
    **for** $t = 0, \ldots, T-1$ **do**
        Perform policy improvement $\mathcal{I}$ with MCTS using $V_\nu(\cdot, R)$ and policy $\pi_\theta(\cdot)$
        Receive improved policy $\mathcal{I}\pi(s_t)$, action $a_t$ and new state $s_{t+1}$
        Store $(s_t, \mathcal{I}\pi(s_t))$ in $\mathcal{M}_\pi$
    Have trajectory $\zeta \leftarrow (s_0, a_0, \ldots, s_{T-1}, a_{T-1}, s_T)$
    $z \leftarrow \begin{cases} 1 & \text{if } r(\zeta) \geq R, \\ -1 & \text{else} \end{cases}$         ▷ outcome as reshaped binary reward
    Store tuples $(s_t, R, z)$ in $\mathcal{M}_V$ for all timesteps $t$

    Periodically update $\theta^B \leftarrow \theta$ if $\sum_{s_0 \in \mathcal{J}_{\text{arena}}} \left( r(\zeta_{0,\pi_\theta}^{\text{greedy}}) - r(\zeta_{0,\pi_{\theta^B}}^{\text{greedy}}) \right) > 0$

---

We set the constants to $c_{\text{visit}} = 50$ and $c_{\text{scale}} = 1.0$, which has shown to be stable across various simulation budgets (Danihelka et al., 2022). We complete the vector of $Q$-values using the authors' value interpolation proposed in their appendix: For every unvisited action, the $Q$-value approximation in (4) is set to $\hat{V}$, where

$$\hat{V} := \frac{1}{1 + \sum_b N(b)} \left( V + \frac{\sum_b N(b)}{\sum_{b, \text{ s.t. } N(b)>0} \pi(b)} \sum_{a, \text{ s.t. } N(a)>0} \pi(a)Q(a) \right). \tag{26}$$

Here, $V$ is the value approximation of node $\mathcal{N}$ coming from the value network. We abuse the notation by omitting states (resp. state pairs for GAZ PTP), as (26) is used in all GAZ variants. In particular, unvisited actions are given zero advantage in (4).

### C.1.3   GAZ NORMALIZATION

Rewards in single-player tasks usually have different scales and are not limited to $\pm 1$. We follow Schrittwieser et al. (2020); Danihelka et al. (2022), and perform a min-max normalization in GAZ Single Vanilla and GAZ Single N-Step (prior to rescaling by $\sigma$) on the vector of completed $Q$-values with the values observed in the tree up to that point. I.e., the maximum value is given by $\max_{s \in \text{tree}} \hat{Q}(s, a)$ (and analogously for the minimum value). This pushes normalized advantages into the interval $[-1, 1]$.

### C.1.4   GAZ GREEDY SCALAR

We provide a listing of the training algorithm for GAZ Greedy Scalar in Algorithm 2.

### C.1.5   GENERAL NETWORK ARCHITECTURE

**Feed-forward** In the following, a feed-forward network (FF) always refers to a multilayer perceptron (MLP) with equal input and output dimensions and one hidden layer of four times the input dimension, with GELU activation (Hendrycks & Gimpel, 2016).

**State encoding** We model a state-encoding network $f \colon \mathcal{S} \to \mathbb{R}^d$, followed by a policy head $g \colon \mathbb{R}^d \to \Delta\mathcal{A}$ and a state-value head $h$, which is of the form $h \colon \mathbb{R}^d \times \mathbb{R}^d \to [-1, 1]$ for GAZ

PTP. For both TSP and JSSP, $f$ is based on the Transformer architecture and its underlying multi-head attention (MHA) layers (Vaswani et al., 2017). For some state $s \in \mathcal{S}$, the network $f$ outputs a concatenation of vectors $f(s) = [\tilde{s}; \tilde{\boldsymbol{a}}_1; \ldots, \tilde{\boldsymbol{a}}_m]$ where $\tilde{s} \in \mathbb{R}^{\tilde{d}}$ and $\tilde{\boldsymbol{a}}_i \in \mathbb{R}^{\tilde{d}}$ are latent representations of the state $s$ and actions $a_1, \ldots, a_m$. The network architecture of $f$ is identical for GAZ PTP and all single-player variants (see C.2.3 and C.3.4).

**Value head**  The value head consists of an MLP with two hidden layers of size $\tilde{d}$ with GELU activation for TSP (and three hidden layers of size $2\tilde{d}$ for JSSP). The input and output of the MLP differ between variants:

- GAZ PTP: We input a concatenation $[\tilde{\boldsymbol{s}}^1; \tilde{\boldsymbol{s}}^{-1}] \in \mathbb{R}^{2\tilde{d}}$ of latent state vectors for both players. The output is mapped to $[-1, 1]$ via $\tanh$-activation.

- GAZ Greedy Scalar: We input a concatenation $[\tilde{\boldsymbol{s}}^1; R] \in \mathbb{R}^{\tilde{d}+1}$, where $R \in \mathbb{R}$ is the outcome of the greedy rollout. As in GAZ PTP, the output is mapped to $[-1, 1]$ via $\tanh$-activation.

- GAZ Single Vanilla/N-Step: Only the latent state $\tilde{s} \in \mathbb{R}^{\tilde{d}}$ serves as the input, with linear output.

**Policy head**  For the policy head $g$, the logit for an action $a_i \in \{a_1, \ldots, a_m\}$ is computed using a pointing mechanism based on the attention of $\tilde{s}$ and the $\tilde{\boldsymbol{a}}_i$'s similarly to (Bello et al., 2016) and (Kool et al., 2018):

We compute (single-head) attention weights $u_1, \ldots, u_m \in [-C, C] \subseteq \mathbb{R}$ via

$$u_i = C \cdot \tanh\left(\frac{(W^Q \tilde{s})^T (W^K \tilde{\boldsymbol{a}}_i)}{\sqrt{\tilde{d}}}\right), \tag{27}$$

with constant $C = 10$ and learnable linear maps $W^Q, W^K \colon \mathbb{R}^{\tilde{d}} \to \mathbb{R}^{\tilde{d}}$. The weight $u_i$ is interpreted as the logit for the probability of action $a_i$ and is set to $-\infty$ for infeasible (masked) actions.

In GAZ PTP, the above process is performed with $\boldsymbol{w} = \mathrm{FF}(\boldsymbol{y}) + \boldsymbol{y}$ instead of $\tilde{s}$, where $\boldsymbol{y} = \mathrm{SHA}(\tilde{\boldsymbol{s}}; \tilde{\boldsymbol{a}}_1, \ldots, \tilde{\boldsymbol{a}}_m))$ is a transformation of the vector $\tilde{s}$ through a layer of single-head attention (SHA) with $\tilde{s}$ as the only query, and keys $\tilde{\boldsymbol{a}}_1, \ldots, \tilde{\boldsymbol{a}}_m$. This is to simplify the task of aligning the state encoding $f$ for the value and policy head, as in GAZ PTP the value head operates on two separate state encodings. The same design choice did not make any difference in GAZ Greedy Scalar/Single Vanilla/N-Step, so we removed it to speed up computation.

### C.1.6  Hyperparameters

Arena episodes are played every 400 episodes in GAZ PTP and GAZ Greedy Scalar. In all experiments, the replay buffer holds data of the latest 2000 episodes. We set the self-play parameter to $\gamma = 0.2$. We use Adam (Kingma & Ba, 2014) as an optimizer, with a constant learning rate of $10^{-4}$, sampling batches of size 256 at each training step. Gradients are clipped to unit $L_2$-norm.

### C.2  TSP

### C.2.1  Environment

An initial state $s_0$ is given by $n$ nodes, where $s_0 = \{\boldsymbol{x}_1, \ldots, \boldsymbol{x}_n\} \subseteq [0, 1]^2 \subseteq \mathbb{R}^2$. An ordered tour is constructed sequentially by picking one node to visit after the other, iteratively completing a partial tour. In particular, actions are represented by (unvisited) nodes. The agent decides from which node it starts the tour.

We take a relative view at timestep $t > 0$ and represent a state (partial tour) for $t > 0$ by a tuple $s_t = (l_t, \boldsymbol{x}_{t,\text{start}}, \boldsymbol{x}_{t,\text{end}}, X_t = \{\boldsymbol{x}_{t_1}, \ldots, \boldsymbol{x}_{t_{n-t}}\})$, where $l_t$ is the length of the current partial tour, $\boldsymbol{x}_{t,\text{start}} = a_{t-1}$ is the last node in the partial tour, $\boldsymbol{x}_{t,\text{end}} = a_0$ is the first chosen node (and must be eventually returned to) and $X_t$ is the remaining set of unvisited nodes from which the next action is picked. The terminal state $s_T$ is a complete solution with $X_T = \emptyset$. The negative length of the full tour is given as a reward at the end of the episode (and zero rewards in between). All tour lengths

are scaled by division with $\sqrt{2}n$ (the supremum of a possible tour length with $n$ nodes in the unit square).

### C.2.2 DATA GENERATION AND TRAINING

Initial states for training are uniformly sampled on the fly. We fix 300 states for the arena set $\mathcal{J}_{\text{arena}}$, which is identical for GAZ PTP and GAZ Greedy Scalar. During training, the model is evaluated periodically on a fixed validation set of 100 states to determine the final model. The final model is evaluated on the 10,000 instances of Kool et al. (2018), which were generated with the random seed 1234. The agent is run for 100k episodes, keeping a ratio of the number of played episodes to the number of optimizer steps of approximately 1 to $0.1n$. The network architecture is independent of the number of input nodes, but for comparability, we train it from scratch for each $n \in \{20, 50, 100\}$. We sample (at most) 16 actions without replacement at the root of GAZ's search tree. The simulation budget in the tree search is 100 for our approach and 200 for all single-player variants.

We augment training data by applying a random reflection, rotation, and linear scaling within the unit square to states sampled from the replay buffer.

### C.2.3 STATE ENCODING NETWORK

The state encoding network $f$ consists of a sequence-to-sequence Transformer architecture similar to the encoder in Kool et al. (2018). We use a latent dimension of $\tilde{d} = 128$ in all TSP experiments. The network is composed of the following components:

- Learnable lookup embeddings $E^{\text{token}}, E^{\text{start}}, E^{\text{end}}, E^{\text{start-ind}}, E^{\text{end-ind}}$ in $\mathbb{R}^{\tilde{d}}$.

- Affine maps $W^{\text{len}}, W^{\text{num}} \colon \mathbb{R} \to \mathbb{R}^{\tilde{d}}$ and $W^{\text{node}} \colon \mathbb{R}^2 \to \mathbb{R}^{\tilde{d}}$.

- A simple stack of five Transformer blocks with eight heads in the self-attention and layer normalization before the MHA and the FF (Wang et al., 2019). The structure of a Transformer block is summarized in Figure 4.

We illustrate the encoding procedure in Figure 4. For an intermediate state $s_t = (l_t, \boldsymbol{x}_{t,\text{start}}, \boldsymbol{x}_{t,\text{end}}, X_t = \{\boldsymbol{x}_{t_1}, \ldots, \boldsymbol{x}_{t_{n-t}}\})$, we construct an input sequence

$$
\begin{aligned}
&(E^{\text{token}}, W^{\text{len}}(l_t), W^{\text{num}}(n - t), \\
&W^{\text{node}}(\boldsymbol{x}_{t,\text{start}}) + E^{\text{start-ind}}, W^{\text{node}}(\boldsymbol{x}_{t,\text{end}}) + E^{\text{end-ind}}, \\
&W^{\text{node}}(\boldsymbol{x}_{t_1}), \ldots, W^{\text{node}}(\boldsymbol{x}_{t_{n-t}})).
\end{aligned}
$$

The sequence element corresponding to $E^{\text{token}}$ is a token representing the state, similar to the class token in natural language processing (NLP) (Devlin et al., 2019). The two-dimensional nodes are affinely embedded into $\mathbb{R}^{\tilde{d}}$. As Transformer architectures are invariant to sequence permutations by design, we add the learnable lookup embeddings $E^{\text{start-ind}}, E^{\text{end-ind}}$ to the start and end nodes to indicate them. This is comparable to position embeddings in NLP. For an initial state $s_0$, we use $W^{\text{len}}(0)$ for the second sequence element. Further, there are no start and end nodes in the initial state yet, so we use the learnable embeddings $E^{\text{start}}, E^{\text{end}}$ instead of the affine embeddings $W^{\text{node}}(\boldsymbol{x}_{t,\text{start}}), W^{\text{node}}(\boldsymbol{x}_{t,\text{end}})$.

The sequence is passed to the stack of Transformer blocks. For each attention head and pair of nodes $\boldsymbol{x}, \boldsymbol{y}$ in the sequence, we add a spatial bias $w_h \cdot \|x - y\|_2 + b_h \in \mathbb{R}$ to the attention weight corresponding to $\boldsymbol{x}, \boldsymbol{y}$, similarly as in Graphormer architectures (Ying et al., 2021).

As outlined in C.1.5, $f$ eventually outputs $f(s_t) = [\tilde{\boldsymbol{s}}; \tilde{\boldsymbol{a}}_1; \ldots, \tilde{\boldsymbol{a}}_{n-t}]$, where $\tilde{\boldsymbol{s}}$ is the first output sequence element corresponding to the state token $E^{\text{token}}$, and $\tilde{\boldsymbol{a}}_i$ corresponds to the output sequence element of the $i$-th remaining node $W^{\text{node}}(\boldsymbol{x}_{t_i})$.

The main structural difference to Kool et al. (2018) is that in our case the latent representation of a state is not computed autoregressively, but is re-done at each state from the partial tour length and two-dimensional coordinates of remaining nodes.

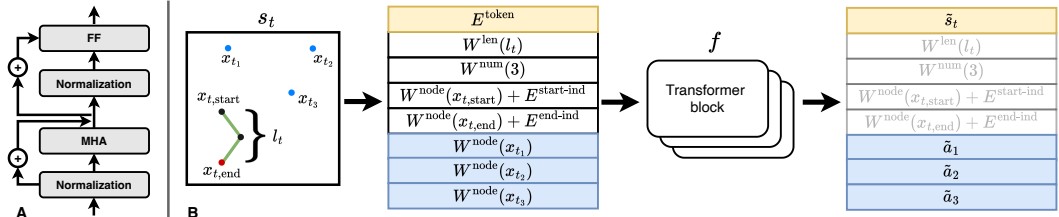

Figure 4: (A) Structure of a Transformer block with pre-normalization. We use layer normalization for TSP. The Transformer block takes in a sequence of elements in $\mathbb{R}^{\tilde{d}}$ and outputs a sequence in $\mathbb{R}^{\tilde{d}}$ of the same length. (B) Schematic view of the relative state encoding for TSP experiments: The agent initially chose to start the tour with the node at the bottom (red). In an intermediate state $s_t$, the last node in the partial tour is the current start node $x_{t,\text{start}}$. The node $x_{t,\text{end}}$ corresponds to the initial node in the tour, as the agent eventually must return to it to complete the tour. A sequence in $\mathbb{R}^{\tilde{d}}$ is constructed from the state and passed through a stack of Transformer blocks. The output sequence elements corresponding to the state token $E^{\text{token}}$ and the unvisited nodes constitute the output of the network $f$.

## C.3  JSSP

### C.3.1  PROBLEM FORMULATION

In the standard JSSP, we are given a set of $k$ jobs $J = \{j_1, \ldots, j_k\}$, each consisting of $m$ operations which need to be scheduled on $m$ machines. Each job $j_i$ is a *permutation* $(o_{i,l})_{l=1}^m$ of the machines, where $o_{i,l} \in \{1, \ldots, m\}$ indicates on which machine the $l$-th operation of job $j_i$ needs to run. Finishing an operation takes some processing time $p_{i,l} \in (0, 1]$. The operations of a job must run in order (precedence constraints), a machine can only process one operation at a time, and there is no preemption. The objective is to find a schedule with minimum makespan.

### C.3.2  ENVIRONMENT

A schedule must satisfy all precedence constraints, and there is a bijection between operations of a job and the $m$ machines. Thus, we can represent a schedule by a (not necessarily unique) sequence of jobs $(\tilde{j}_1, \ldots, \tilde{j}_{k \cdot m})$, where $\tilde{j}_i \in J$ is an unfinished job of which the next unscheduled operation should be scheduled at the earliest time possible. In particular, we can represent feasible actions in the environment by the set of unfinished jobs, limiting the number of possible actions to $k$. Note that a timestep $t$ in the environment corresponds to the $t$-th chosen action (unfinished job), and is not equal to the passed processing time in the schedule.

A state $s_t$ is given by a tuple

$$((c_{t,l})_{l=1}^m, (e_{t,i})_{i=1}^k, J_t),$$

where

- $c_{t,l} \in \mathbb{R}_{\geq 0}$ is the finishing time of the latest scheduled operation on the $l$-th machine ('machine availability'),

- $e_{t,i} \in \mathbb{R}_{\geq 0}$ is the finishing time of the last scheduled operation of a job $j_i$ ('job availability'), and

- $J_t = \{j_{t,1}, \ldots, j_{t,n_t}\} \subseteq J$ is the subset of jobs with unscheduled operations ('unfinished jobs').

In particular, we have $c_{0,l} = 0$, $e_{0,i} = 0$ and $J_0 = J$ at the initial state $s_0$. Let $\tilde{c}_t := \min_l c_{t,l}$ in the following. The negative makespan $\max_l c_{t,l}$ is given as a reward at the end of the episode (and zero rewards in between). Similarly to the TSP environment, we scale the final makespan by division with 100, so that for all problem sizes the makespan lies roughly in $(0, 1)$. We illustrate the state representation in Figure 5.

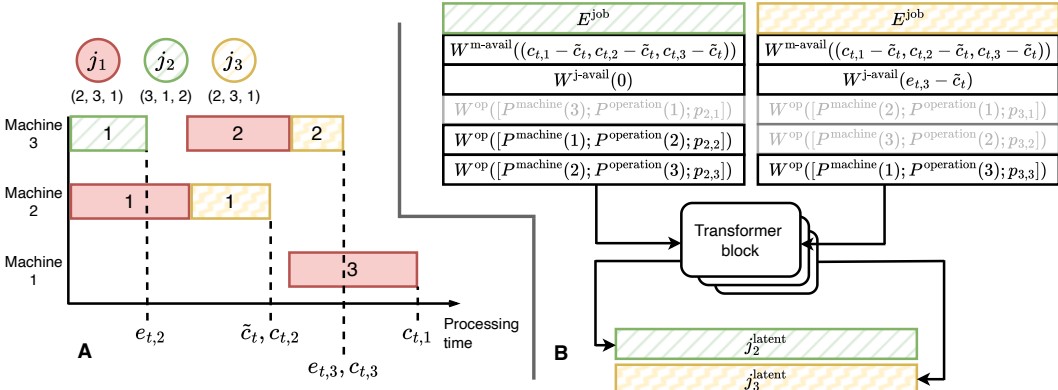

Figure 5: (A) Exemplary Gantt chart of an unfinished schedule, illustrating the relative state representation for JSSP: We assume an instance with three jobs $j_1, j_2, j_3$ and three machines. The tuple below a job indicates on which machines its operations must run. In the depicted unfinished schedule, all three operations of $j_1$, the first operation of $j_2$, and the first two operations of $j_3$ have been scheduled. The schedule can be represented by the sequence $(j_2, j_1, j_1, j_1, j_3, j_3)$. Dashed lines indicate examples of machine and job availability times. (B) Latent representations for the unfinished jobs $j_2, j_3$ are obtained by passing a sequence of operations to the first stack of Transformer blocks. Already scheduled operations are masked. The ouput tokens corresponding to $E^{\text{token}}$ are used as latent job representations and form part of the sequence on which a second Transformer network operates.

Table 2: Best solutions for Taillard instances from the literature, "*" means the solution is optimal.

| Ta01 | Ta02 | Ta03 | Ta04 | Ta05 | Ta06 | Ta07 | Ta08 | Ta09 | Ta10 |
|------|------|------|------|------|------|------|------|------|------|
| 1231* | 1244* | 1218* | 1175* | 1224* | 1238* | 1227* | 1217* | 1274* | 1241* |

| Ta21 | Ta22 | Ta23 | Ta24 | Ta25 | Ta26 | Ta27 | Ta28 | Ta29 | Ta30 |
|------|------|------|------|------|------|------|------|------|------|
| 1642* | 1600 | 1557 | 1644* | 1595 | 1643 | 1680 | 1603* | 1625 | 1584 |

| Ta41 | Ta42 | Ta43 | Ta44 | Ta45 | Ta46 | Ta47 | Ta48 | Ta49 | Ta50 |
|------|------|------|------|------|------|------|------|------|------|
| 2005 | 1937 | 1846 | 1979 | 2000 | 2006 | 1889 | 1937 | 1961 | 1923 |

### C.3.3 DATA GENERATION AND TRAINING

For a given number of jobs $k$ and machines $m$, we generate training data on the fly by randomly sampling a machine permutation $(o_{i,l})_{l=1}^m \in S_m$ for each job $j_i$, and a random processing time $p_{il} \in (0, 1]$ for each operation. We fix 200 instances for $\mathcal{J}_{\text{arena}}$ and a small validation set of size 20. We evaluate the model on the 10 benchmark instances of Taillard (Taillard, 1993) for each size: instances ta01-ta10 (size $15 \times 15$), instances ta21-ta30 (size $20 \times 20$) and instances ta41-ta50 (size $30 \times 20$). The benchmark instances have integer processing times in $[1, 100]$, which we rescale to the unit interval by division with 100. We summarize the best upper bounds from the literature in Table 2, as reported in Zhang et al. (2020). An episode takes $k \cdot m$ actions, so to reduce computation time, we limit the simulation budget to 50 for our approach and 100 for all single-player variants, running the agent for 20k episodes. We keep a ratio of the number of played episodes to the number of optimizer steps of approximately 1 to $0.02k \cdot m$. We train from scratch for each problem size. As the action space is rather small (at most $k$ actions at each timestep), we consider *all* feasible actions for the simulations at the root of the search tree.

During training, we augment states sampled from the replay buffer by linearly scaling processing times $p_{i,l}$, job availability times $e_{t,i}$ and machine availability times $c_{t,l}$ with a random scalar in $(0, 1)$. Furthermore, we shuffle the machines on which operations must be scheduled (e.g. operations on some machine $A$ are reassigned to some machine $B$ and the other way round).

### C.3.4 STATE ENCODING NETWORK

There are two types of sequences in a problem instance. (i) For each job $j_i$, we have a sequence of operations, where the order of operations matters. (ii) The entirety of jobs forms a sequence, where the order does *not* matter (similar to the sequence of nodes in the TSP). The encoding network $f$ consists of two stacked Transformer models, where the first one computes a latent representation for each job separately, and the second one operates on the sequence of these job representations to compute a state encoding. Both networks operate in a latent space of dimension $\tilde{d} = 64$.

**Job encoding**  The Transformer network for encoding each job from the sequence of its unfinished operations consists of:

- Learnable one-dimensional embeddings $P^{\text{machine}}, P^{\text{operation}} \colon \{1, \ldots, m\} \to \mathbb{R}^{\tilde{d}}$.
- Learnable lookup embedding $E^{\text{job}}$ in $\mathbb{R}^{\tilde{d}}$.
- Affine maps $W^{\text{op}} \colon \mathbb{R}^{2\tilde{d}+1} \to \mathbb{R}^{\tilde{d}}$, $W^{\text{m-avail}} \colon \mathbb{R}^m \to \mathbb{R}^{\tilde{d}}$, and $W^{\text{j-avail}} \colon \mathbb{R} \to \mathbb{R}^{\tilde{d}}$.
- A simple stack of three Transformer blocks with four heads in the self-attention and instance normalization before the MHA and the FF.

Let $j_i \in J_t$ be an unfinished job. We construct a sequence in $\mathbb{R}^{\tilde{d}}$

$$
\begin{aligned}
(&E^{\text{job}}, \\
&W^{\text{m-avail}}((c_{t,1} - \tilde{c}_t, \ldots, c_{t,m} - \tilde{c}_t)), W^{\text{j-avail}}(\max\{0, e_{t,i} - \tilde{c}_t\}), \\
&W^{\text{op}}([P^{\text{machine}}(o_{i,1}); P^{\text{operation}}(1); p_{i,1}]), \\
&\quad \vdots \\
&W^{\text{op}}([P^{\text{machine}}(o_{i,m}); P^{\text{operation}}(m); p_{i,m}]),
\end{aligned}
$$

which is passed to the stack of Transformer blocks. We mask all operations which have already been scheduled. The first element in the output sequence corresponding to $E^{\text{job}}$ is taken as a latent representation of the job $j_i$, which we denote by $j_i^{\text{latent}} \in \mathbb{R}^{\tilde{d}}$.

The job encoding procedure is depicted in Figure 5.

**State encoding**  The network for encoding the state representation from the sequence of latent job representations consists of:

- Learnable lookup embedding $E^{\text{token}}$ in $\mathbb{R}^{\tilde{d}}$.
- Affine maps $W^{\text{num}}, W^{\text{len}} \colon \mathbb{R} \to \mathbb{R}^{\tilde{d}}$, $W^{\text{job}} \colon \mathbb{R}^{\tilde{d}} \to \mathbb{R}^{\tilde{d}}$, and $W^{\text{m-avail-2}} \colon \mathbb{R}^m \to \mathbb{R}^{\tilde{d}}$.
- A simple stack of four Transformer blocks with four heads in the self-attention and instance normalization before the MHA and the FF.

From the set of unfinished jobs $J_t = \{j_{t,1}, \ldots, j_{t,n_t}\}$, we construct a sequence

$$
\begin{aligned}
(&E^{\text{token}}, \\
&W^{\text{num}}(n_t), W^{\text{len}}(\tilde{c}_t), W^{\text{m-avail-2}}((c_{t,1} - \tilde{c}_t, \ldots, c_{t,m} - \tilde{c}_t)), \\
&W^{\text{job}}(j_{t,1}^{\text{latent}}), \cdots, W^{\text{job}}(j_{t,n_t}^{\text{latent}})),
\end{aligned}
$$

which is passed to the stack of Transformer blocks. Finally, $f$ outputs $f(s_t) = [\tilde{s}; \tilde{a}_1; \ldots, \tilde{a}_{n-t}]$, where $\tilde{s}$ is the first output sequence element corresponding to $E^{\text{token}}$, and $\tilde{a}_i$ corresponds to the output sequence element of the $i$-th unfinished job $W^{\text{job}}(j_{t,i}^{\text{latent}})$.

## D  VALUE ESTIMATES AS TIMESTEP-DEPENDENT BASELINES

In this section, we provide more insight into why comparing pairs of *states* in the value function instead of directly comparing pairs of *predicted values* can be beneficial in a self-competitive setting.

To recall, at some state $s_t$ for an action $a_t$, the policy logit update in GAZ (both for the in-tree action selection and obtaining a policy training target) is given by

$$\text{logit}^{\pi'_{\text{GAZ}}}(a) = \text{logit}^\pi(a) + \sigma(\hat{A}^\pi(s, a)),$$

where $\hat{A}^\pi(s, a) = \hat{Q}^\pi(s_t, a_t) - \hat{V}^\pi(s_t)$ is an advantage estimation based on the $Q$-value estimates from the tree search and value network evaluations (cf. equation (4) in Section 3.2). In the following discussion, for ease of notation, we denote by $\pi := \pi_\theta$ the current policy of the learning actor, and by $\mu := \pi_{\theta^B}$ the historical best greedy policy.

As proposed in Section 3.2, in GAZ PTP, we are comparing *pairs of states* at a timestep $t$ to assess how good the learning actor performs compared with its historical version. This is achieved by swapping the term $\hat{A}^\pi(s_t, a_t)$ in the logit update with

$$\hat{A}^{\pi,\mu}_{\text{sgn}}(s_t, s'_t; a_t) := \hat{Q}^{\pi,\mu}_{\text{sgn}}(s_t, s'_t; a_t) - \hat{V}^{\pi,\mu}_{\text{sgn}}(s_t, s'_t),$$

where the state $s'_t$ comes from a (greedy) trajectory of the policy $\mu$ (see equation (7)). The sgn in the subscript indicates that the value network is trained to estimate the sign of the episodic reward difference in the original MDP (cf. equation (6)). In the self-competitive framework, we are working with the assumption that predicting the expected episode outcome without sophisticated techniques can be a hard task. By supplying the value network with training data consisting of state pairs and binary targets to decide which state is more advantageous, the network is given a certain amount of freedom in how to model the problem and does not rely explicitly (as e.g. GAZ Single Vanilla) on the value network's capability to predict the expected outcome.

In contrast, by explicitly using value predictions, the advantage estimation at a timestep $t$ can also be formulated as

$$\hat{A}^{\pi,\mu}_t(s_t, a_t) := \hat{Q}^\pi(s_t, a_t) - b^\mu_t, \tag{28}$$

where the baseline $b^\mu_t$ is an estimate of $\mathbb{E}_{\zeta_0=(s_0, a'_0, s'_1, \ldots, s'_T) \sim \eta^\mu(s_0)} [V^\mu(s'_t)]$. In this case, we are baselining the $Q$-values from the standard single-player tree search with the expectation of the *value* of the historical policy $\mu$ at timestep $t$. This is sensible, as it allows us to compute the estimate $b^\mu_t$ for all timesteps in advance of the learning actor's episode (similarly to GAZ PTP GT, see B.6) without the need to reevaluate the states $s'_i$ encountered by $\mu$ in the tree search. By using $b^\mu_t$, we maintain the postulated benefit of keeping instance-specific information about the behavior of $\mu$ in intermediate timesteps (via value estimates). Additionally, the MCTS can be run with GAZ in the standard single-player way, as baselining with $b^\mu_t$ fits elegantly into GAZ's in-tree action selection and policy improvement mechanisms (see equations (4) and (28)). In the following, we refer to this method as GAZ Single Timestep Baseline (GAZ Single TB) and consider two ways of obtaining $b^\mu_t$:

(i) (**GAZ Single TB Greedy**) Let $\zeta^{\text{greedy}}_0 = (s_0, a'_0, \ldots, s'_{T-1}, a'_{T-1}, s'_T)$ be the trajectory obtained by rolling out $\mu$ *greedily*. As a counterpart to GAZ PTP GT, we set

$$b^\mu_t := V_\nu(s'_t),$$

where $V_\nu: \mathcal{S} \to \mathbb{R}$ is the learning actor's (single-player) value network.

(ii) (**GAZ Single TB Sampled**) For $i \in \{1, \ldots, k\}$ for some $k \in \mathbb{N}$, we *sample* trajectories

$$\zeta_{0,i} = (s_0, a'_{0,i}, \ldots, s'_{T-1,i}, a'_{T-1,i}, s'_{T,i})$$

using $\mu$, and average over the value network evaluations via

$$b^\mu_t := \frac{1}{k} \sum_{i=1}^k V_\nu(s'_{t,i}).$$

We provide a listing of the method in Algorithm 3.

The algorithmic variant GAZ PTP GT is similar to GAZ Single TB Greedy, except that we compare *pairs of states* in GAZ PTP GT, and *value estimates* in GAZ Single TB Greedy. Furthermore, the number of network evaluations is the same in both variants because only a single greedy rollout of $\mu$ is required. Value estimates (resp. latent states for GAZ PTP GT) can be stored in memory for usage in the MCTS. We compare the two variants of GAZ Single TB with GAZ Single Vanilla and GAZ PTP GT in small-scale experiments for TSP 100 (50 search simulations, 20k episodes) and JSSP

---

**Algorithm 3:** GAZ Single TB Training

---

**Input:** $\rho_0$: initial state distribution
**Input:** $\mathcal{J}_{\text{arena}}$: set of initial states sampled from $\rho_0$
Init policy replay buffer $\mathcal{M}_\pi \leftarrow \emptyset$ and value replay buffer $\mathcal{M}_V \leftarrow \emptyset$
Init parameters $\theta, \nu$ for policy net $\pi_\theta : \mathcal{S} \rightarrow \Delta\mathcal{A}$ and value net $V_\nu : \mathcal{S} \rightarrow \mathbb{R}$
Init 'best' parameters $\theta^B \leftarrow \theta$
**foreach** episode **do**

    Sample initial state $s_0 \sim \rho_0$
    Obtain $T$ baseline values $b_0, \ldots, b_{T-1}$ via

$$b_t \leftarrow \begin{cases} V_\nu(s'_t) & \text{for greedy trajectory } \zeta_0^{\text{greedy}} \quad \triangleright \text{ TB Greedy} \\ \frac{1}{k} \sum_{i=1}^{k} V_\nu(s'_{t,i}) & \text{for } k \text{ sampled trajectories } \zeta_{0,i} \quad \triangleright \text{ TB Sampled} \end{cases}$$

    **for** $t = 0, \ldots, T-1$ **do**
        Perform policy improvement $\mathcal{I}$ with single-player MCTS using $V_\nu(\cdot)$ and policy $\pi_\theta(\cdot)$,
            baselining logit updates with $b_t, \ldots, b_{T-1}$ in tree
        Receive improved policy $\mathcal{I}\pi(s_t)$, action $a_t$ and new state $s_{t+1}$
    Have trajectory $\zeta \leftarrow (s_0, a_0, \ldots, s_{T-1}, a_{T-1}, s_T)$
    Store $(s_t, \mathcal{I}\pi(s_t))$ in $\mathcal{M}_\pi$ for all timesteps $t$
    Store $(\zeta, r(\zeta))$ in $\mathcal{M}_V$

    Periodically update $\theta^B \leftarrow \theta$ if $\sum_{s_0 \in \mathcal{J}_{\text{arena}}} \left( r(\zeta_{0,\pi_\theta}^{\text{greedy}}) - r(\zeta_{0,\pi_{\theta^B}}^{\text{greedy}}) \right) > 0$

---

Table 3: Results for TSP $n = 100$ (50 simulations and 20k episodes) and JSSP $15 \times 15$ (35 simulations and 10k episodes). Results are averaged $\pm$ standard deviation across three seeds $\in \{42, 43, 44\}$. GAZ Single Vanilla fails to learn for seed 44.

| | TSP $n = 100$ | | JSSP $15 \times 15$ | |
|---|---|---|---|---|
| Method | Obj. | Gap | Obj. | Gap |
| GAZ PTP GT | $8.06 \pm 0.04$ | $3.8\% \pm 0.5\%$ | $1505.2 \pm 55.7$ | $22.0\% \pm 3.8\%$ |
| GAZ Single Vanilla | $10.76 \pm 0.04$ | $38.6\% \pm 0.5\%$ | $2823.6 \pm 1787.5$ | $129.8\% \pm 145.4\%$ |
| GAZ Single TB Greedy | $10.30 \pm 0.19$ | $32.6\% \pm 2.5\%$ | $1532.5 \pm 20.3$ | $24.7\% \pm 1.6\%$ |
| GAZ Single TB Sampled | $10.82 \pm 0.20$ | $39.4\% \pm 2.6\%$ | $1910.8 \pm 498.3$ | $55.5\% \pm 40.6\%$ |

$15 \times 15$ (35 search simulations, 10k episodes). For GAZ Single TB Sampled, we sample $k = 10$ trajectories. The results are presented in Table 3.

Even though GAZ Single TB still relies on the value network being able to predict the expected outcome of an episode sufficiently well, the small-scale experiments indicate that GAZ Single TB provides a better advantage baseline to form a curriculum for the learning actor than the value interpolation of GAZ Single Vanilla. GAZ Single TB Greedy in particular obtains comparable results to GAZ PTP GT for JSSP, and improves faster than GAZ Single Vanilla for TSP. We believe that GAZ Single TB can provide a much stronger method than the value interpolation (26) of GAZ for baselining the Q-values in problems where it is 'easier' to predict the expected outcome of an episode, or in combination with more sophisticated value prediction techniques, such as the target scaling techniques proposed in Pohlen et al. (2018) or Implicit Quantile Networks (Dabney et al., 2018).

