# OpenReview forum: "Policy-Based Self-Competition for Planning Problems"
_ICLR.cc/2023/Conference — ICLR 2023 poster_

### Official Review · Reviewer_T4pL · 2022-10-23

**Confidence:** 4
**Correctness:** 4
**Technical Novelty And Significance:** 3
**Empirical Novelty And Significance:** 3
**Recommendation:** 8

**Clarity, Quality, Novelty And Reproducibility:**

**Quality** / **Novelty** / **Originality**: looks good

**Reproducibility**: probably sufficient detail provided, but no mention of source code (please correct me if I missed it)

**Clarity**: my main concern is here. At least in some way. I do think the writing is good overall and the contributions and experiments etc. are clearly described. There are just some design decisions in the proposed approach, for which I'm missing any motivation (intuition or otherwise):

## Why the 2-player game model?

This is the main question that I keep wondering about. On one hand, I do think the "trick" where the single-agent problem is rephrased as a kind of 2-player game is very interesting, and cool, and I understand how the general model works. On the other hand, I'm just not convinced that it's actually necessary, at least not in its entirety, and can't help but keep thinking that it's unnecessarily complicated (which may be a detriment to the efficiency of the algorithm itself, as well as the clarity of the paper).

So, first, here are the parts that I *do* understand and follow:
- I understand the idea behind self-critical learning / self-competition, and that you want some baseline value to measure progress against, both to form a natural curriculum and to avoid the need for reward normalisation/scaling.
- I understand that it may be better not to determine this baseline value by running a complete rollout and summarising it as a scalar, since this causes us to lose a bunch of information.

Now, here is roughly where I suspect that my thoughts/intuition start to deviate from the authors':
- I do not think it is necessary (or useful) to ever compare **states** that the "main" player is in, to **states** that the greedy/rollout/baseline actor is in. These two should never influence each other. For sure, the greedy actor never observes anything involving the main player. So, the entire rollout for the greedy actor could, in principle, be executed in its entirety first (and stored in memory, rather than summarised as a scalar). There is no need to run two trajectories for the two players simultaneously. As far as I can tell, the main player should also never look at the states that the greedy actor was in. According to the model as described in the paper, the main player is *allowed* to, but I do not see why it would ever be *useful*.
- Rather than comparing pairs of **states**, I think all the involved functions (V, Q, A) should only be based on comparing states of the main player to **time steps** of the greedy actor. I.e.: is the main actor, in some state $s_t$ at some time step $t$, in a better or a worse state than the greedy actor is *in expectation* at the same time step $t$? A positive or a negative reward can be given accordingly. We should not care about whether we are in a better or worse state than the state the greedy actor happened to be in during a single trajectory that we happened to run simultaneously, because they should be completely independent. We should care much more about the value of our current state relative to the expected value of the greedy actor at the same point in time.
- Following from the previous two points I think it is harmful to actually model the tree search as a 2-player game, with explicit nodes and branches for the greedy actor in its completely-independent-but-simultaneously-running other game. As far as I am able to understand, this just unnecessarily blows up (exponentially) the size of the search tree. There will be many many different nodes in the search tree, with identical sequences of move selections by the main player but different sequences of move selections by the greedy player, which really should all just be the same node with respect to the original single-player model, but are unnecessarily treated as distinct situations in the tree search.
- For the main player, it would seem much more sensible to me to keep just a search tree for its own single-agent game. However, prior to running the episode for the search-based main player, I would generate a bunch (e.g., 100) of rollouts for the greedy (actually sampling) agent. The particular states it encounters are irrelevant, but the estimated values (as predicted by value function evaluations) can be stored per time step (averaged over the multiple rollouts) to get an estimate of how good the greedy agent would do at every time step. Then, these can be used, for every appropriate time step, as time-step-dependent baselines, relative to which rewards can be assigned to the main agent.
- I think that what I just described is somewhat similar to **GAZ Greedy Scalar** as described on page 7, except it would (1) probably run multiple rollouts instead of just 1, to average over, (2) sample instead of playing greedily, as in your main approach, and (3) track per-timestep baselines from value function evaluations, instead of summarising everything into a single scalar.

---

## Minor comments
- Bottom of p.2: Mentions AlphaZero, but provides reference for AlphaGo (which actually did still run full rollouts).
- For the (Schadd et al.) reference, no year of publication and no title is provided

**Strength And Weaknesses:**

**Strengths**:
- Contributions look sound, interesting ideas, also novel
- Promising/good empirical results

**Weaknesses**:
- Some of the core ideas behind the approach leave me wondering about a bunch of unaddressed questions. I think these need to be cleared up somehow. See detailed comments below.

---

**After discussion with authors**: The authors' comments and added content in the new revision have helped me a lot to clear up confusion.

**Summary Of The Paper:**

This paper describes a new "policy-based self-competition" approach for training policies for single-agent (planning) problems with Gumbel AlphaZero. It uses rollouts from a DNN policy (without search) to provide reference/baseline values, such that binary reward signals (without any need for normalisation) for the DNN+search agent can be provided depending on whether or not it outperforms the baseline.

**Summary Of The Review:**

An interesting paper with a lot of potential, but there is a major point where I can't help but feel like the problem is modelled in a way that is, unnecessarily, hugely overcomplicated (even if it is a very interesting way to model it). If I am correct, I feel that this is an important issue, hence my current score. But I also do not rule out the possibility that I simply completely missed some key detail, in which case I hope the authors will be able to provide clarification.

---

**After discussion with authors**: The authors' comments and added content in the new revision have helped me a lot to clear up confusion.

---

> ### Author Response · Authors · 2022-11-16
> **Response to Reviewer T4pL (Part 1)**
>
> Dear reviewer,
>
> Thank you for taking the time to read and review our paper. We very much appreciate the constructive and detailed feedback and that you believe that our paper has a lot of potential.
> We also appreciate that you have shared your concerns about the method in much detail.
> There is a lot to unpack, so we structure our answer as follows (please excuse that it responds to some of your points not in the original order):
>
> First, we answer the question “Why compare pairs of states and not estimated values?” addressed in your first two points, and briefly touch on the timestep-dependent method you proposed in your last two points. Afterward, we discuss the harmfulness/unnecessary complexity of modeling the tree search as a 2-player game due to the blowup of the search tree (addressed in your third point). Finally, we return to the timestep-dependent method and discuss additional experiments performed with this method.
>
> We indicate changes made to the manuscript in **bold**.
>
> ### Why should we compare states at similar timesteps and not estimated values?
>
> This question is important. To clarify it as best as possible, please allow us to go back a little:
> As an opening statement, the ‘regular’ GAZ single-player variant does work perfectly fine if we can estimate the value function sufficiently well. If we had perfect knowledge of the value function, the whole self-competitive/self-critical methodology is questionable because it does not lead to any advantage at all. However, we know that this simplification is usually not valid in practice, since value function approximations can be notoriously hard. E.g., even small changes such as linear reward scaling can have a strong positive/negative impact on an approximator’s performance [3]. There is a richness of work designing methods for/around the value approximation problem. For example, let’s think of MuZero in Atari, where the value target is represented as a categorical problem with 601 categories (following [4]).
> All in all, the picture changes as soon as we assume that a good value function approximator is hard, complicated and/or fiddly to obtain, which is a major motivation of self-critical training (where it is avoided to learn a value function at all).
>
> So in the following, we suppose that the value function is hard to estimate, and let’s additionally suppose that we use the method you sketched out with timestep-dependent baselines coming from value estimates (which we think is a good idea. For this purpose, we added a new section “D. Value estimates as timestep-dependent baselines” to the appendix of our manuscript in which we implemented, tested, and discussed it. We will come back to this in detail below).
> Then, as you mentioned, the baselines are suitable because they maintain the postulated benefit of keeping instance-specific information about the behavior of the historical policy in intermediate timesteps.
> *But:* the baseline still relies on the value function approximation to be sufficiently exact (even when taking the mean of the estimates), which can be hard and has been exactly one of the reasons to switch to a self-competitive framework in the first place.
> So as we are doing (greedy) rollouts as in self-critical training (where baselining with the outcome of a single trajectory has proven quite successful), we propose to baseline the modified Q-values at a timestep $t$ with the state-comparing function $V^{\pi, \mu}_{\text{sgn}}(s_t, s_t’)$ (as defined in eq. (6)), because it *uncouples* the value network from the explicit requirement to predict the expected outcome (i.e., final episodic reward in our setup) from a state. Furthermore, by training the network on pairs of states with binary targets, it is given a certain amount of freedom in how the network models the problem and decides if the learning actor is in a more advantageous position than its historical counterpart. Certainly, in theory, we have the equality $V^{\pi, \mu}(s_t, s_t’) = V^{\pi}(s_t) - V^{\mu}(s_t’)$ (see proof of Lemma 1 in Appendix A), but for estimating the sign of the value difference $V^{\pi, \mu}_\text{sgn}(s_t, s_t’)$, the network is not forced to model its decision internally via the (hard to predict) explicit difference $V^{\pi}(s_t) - V^{\mu}(s_t’)$.

---

> > ### Author Response · Authors · 2022-11-16
> > **Response to Reviewer T4pL (Part 2)**
> >
> > Some pieces of evidence that it is indeed not modeled internally in this way are given by the results of the variant GAZ Greedy Scalar (see Table 1). Here, the value net is trained on binary targets as well but needs to compare a state against the explicit scalar outcome of the greedy rollout. The method works in some cases but generally underperforms. One could argue that the greedy rollout does not properly reflect $V^\mu(s_t’)$. However, this does not change the general dynamics, as we will see later. To conclude, if you would allow an informal comparison with the ‘intuition’ we mentioned in the introduction concerning board states in a game: It’s like a chess player who might be able to tell from a board configuration “White will likely win this game”, but might not be able to say “White will likely win this game with 3 pawns ahead.”
> >
> > One additional remark: The policy depends as usual just on one state. The comparison of pairs of states affects exclusively the value function with outputs in [-1, 1] and guides the search. So the main player is not only *allowed* to look at the opponent’s state, it *must* look at the state to assess its current performance.
> >
> > Before we discuss in detail the timestep-dependent baseline method you proposed, we would like to address your third point regarding the harmfulness of modeling the tree search as a 2-player game.
> >
> > > Following from the previous two points I think it is harmful to actually model the tree search as a 2-player game, with explicit nodes and branches for the greedy actor in its completely-independent-but-simultaneously-running other game. As far as I am able to understand, this just unnecessarily blows up (exponentially) the size of the search tree.
> >
> > We agree that this blows up the size of the search tree, as mentioned in the “Limitations” paragraph in Section 5.3. It’s also correct that identical actions of the main player can lead to different branches in the tree by sampling actions for the greedy actor. In later stages of training, this branching reduces when the action distribution of the policy gets sharper. For example, for TSP n=100, the highest action probability of the greedy actor’s final policy has a probability of ~0.14 for the first move (when there are all 100 cities left) and ~0.8 after 20 moves, so these actions will get sampled over and over again. Furthermore, the empirical results in Table 1 and Figure 1 (former Figure 2) show that the in-tree sampling leads to finding strong trajectories.
> >
> > Nevertheless, we think you have raised a valid point that needs further examination. Due to this, we included a GAZ PTP variant in the paper, where the moves of the greedy actor are *also chosen greedily in the search tree*. Everything else in the game dynamics stays the same, with Algorithm 1 being a common description of both variants. Thus, we obtain two variants of our method, GAZ PTP ST (“sampled tree”) and GAZ PTP GT (“greedy tree”). **We introduce and highlight the pros/cons of both methods in the paragraph “(i) Choice of actions” in Section 4.2.** In summary, these are:
> >
> > - When *sampling* the actions of the greedy player, exploration is encouraged (as other non-greedy trajectories of the greedy actor are considered), but (as you mentioned) identical actions can lead to different branches.
> > - When choosing the actions *greedily*, this is computationally much more efficient, as only the greedy trajectory of the greedy actor is considered by the main player throughout the tree search. In particular, this variant can be implemented efficiently by executing the greedy trajectory entirely first (as you are also suggesting) and storing *latent states* in memory which are then in the tree search compared to the latent states of the main player (because in practice the network has a latent state representation part and a policy and value head). By this, we maintain a single network evaluation per search simulation, as desired. **In the main text, we added a reference in the paragraph above to an added section “B.6 Efficient implementation of GAZ PTP GT” in the appendix which makes this notion explicit.** The downside of this method is that (as we are only considering one trajectory), the main player might improve slowly if the historical policy is too weak or too strong.
> >
> > In the following, we refer to GAZ PTP ST as “ST” and GAZ PTP GT as “GT”.
> > **We added experiments for GT and its analysis to the experimental section.** In summary: The results in Table 1 show that GT yields consistently better results than ST for TSP and are comparable to ST for JSSP (with GT being generally more favorable due to being computationally more efficient). The seed experiments in Figure 1 (former Figure 2) show that ST improves faster at the beginning, which hints at the improved exploration, and that GT needs more time until the curve 'drops'.

---

> > > ### Author Response · Authors · 2022-11-16
> > > **Response to Reviewer T4pL (Part 3)**
> > >
> > > We sincerely thank you for moving this more into focus. The inclusion of both variants improves the paper a lot, as the comparison sheds more light on the sampling method during the search. All in all, both methods have their right to exist due to their pros and cons and because both stem from the same theoretical motivation. Hence, our main contribution as summarized in Algorithm 1 still stands for both variants as a general description, even if GT can (and should) be implemented more efficiently.
> > >
> > > We now have everything to discuss the timestep-dependent baselining method that you proposed. First of all, thank you for going so much into detail in your thoughts. For us as authors, it is a pleasure to discuss these ideas.
> > >
> > > We really like the idea, as it also addresses the goals of our method (maintaining instance-specific information about intermediate timesteps of the greedy actor, as mentioned above). Furthermore, with a slight variation (but maybe you also meant exactly this), your proposed method fits elegantly into the Gumbel AlphaZero action-selection mechanism:
> > > The main player uses GAZ in the ‘usual’ single-agent way and obtains rewards according to the original MDP (in our case, the objective given at the end of the episode). The timestep-dependent baselines are used to baseline the ‘regular’ Q-values in the advantage function of GAZ’s logit update (which is used both for the in-tree action selection and for computing the policy target).
> > >
> > > As this method provides good grounds for discussing the points above concerning “Why comparing states?”, **we added an entire section dedicated to this in the appendix “D. Value estimates as timestep-dependent baselines”.** In this additional section, we formalized the idea, termed it GAZ Single TB (‘timestep baseline’), and implemented two variants of it:
> > >
> > > - GAZ Single TB *Greedy*: At timestep $t$, the value estimate of $V(s’_t)$ (coming from the value network) of the state $s’_t$ encountered in the greedy trajectory of the greedy actor is directly used as a baseline.
> > > - GAZ Single TB *Sampled*: (As you proposed) The mean over the value estimates at states at timestep $t$ from sampled trajectories is used.
> > >
> > > Note that GAZ Single TB Greedy is similar to GAZ PTP GT (the variant of our method which also uses only the greedy rollout), except that we compare *value estimates* in one and pairs of states in the other. Hence, we conducted small-scale experiments with three seeds on TSP n=100 and JSSP 15x15 and compared GAZ Single Vanilla (‘regular’ single-player GAZ), GAZ PTP GT (ours), and the two GAZ Single TB approaches above. Due to time constraints, we sampled ten trajectories for the sampling method. To summarize the results in Table 3 (we copy the table in a comment below for easier reference):
> > >
> > > We note that GAZ Single TB still requires the value function approximation to work sufficiently well and does not reach the performance of GAZ PTP GT (ours). But GAZ Single TB Greedy and GAZ PTP GT are almost on-par on JSSP 15x15. Furthermore, both GAZ Single TB variants perform far better on JSSP and improve faster on TSP than GAZ Single Vanilla (the ‘regular’ single-player GAZ). We conclude that GAZ Single TB can be a stronger self-competitive method than the value interpolation of GAZ (as in eq. (26)) for baselining the Q-values in problems where it is ‘easier’ to predict the expected outcome or in combination with more sophisticated value prediction techniques. However, the benefit of GAZ PTP GT in the experiments lies in the pairwise state comparison as discussed in the beginning of our response.
> > > Thank you for bringing this to our attention, we think the discussion in Appendix D improves the paper a lot, and **we refer to Appendix D (with full details and discussion) in the paragraph “Value estimates as baselines” in Section 5.3.**
> > >
> > > ### Minor comments
> > >
> > > Thank you for pointing out the reference errors. They are fixed in the current manuscript. Regarding the source code, we added a link to an anonymous repository in a general comment when the discussion forum opened: https://anonymous.4open.science/r/GAZ-PTP-2A0F
> > > We will add the source code for the GAZ Single TB method shortly.
> > >
> > > Thank you for reading our paper and for the insightful feedback and discussion. It helped us a lot to refine our paper! Please let us know if there are additional questions.
> > >
> > > ---
> > >
> > > [3] Henderson, Peter, et al. "Deep reinforcement learning that matters." Proceedings of the AAAI conference on artificial intelligence. Vol. 32. No. 1. 2018.
> > > [4] Pohlen, Tobias, et al. "Observe and look further: Achieving consistent performance on atari." arXiv preprint arXiv:1805.11593 (2018).

---

> > > > ### Author Response · Authors · 2022-11-16
> > > > **Response to Reviewer T4pL (Part 4): Results table**
> > > >
> > > > Results for TSP $n$ = 100 (50 simulations and 20k episodes) and JSSP $15 \times 15$ (35 simulations and 10k episodes). Results are averaged $\pm$ standard deviation across three seeds $\in \{42, 43, 44\}$. GAZ Single Vanilla fails to learn for seed 44.
> > > >
> > > > | | TSP $n$ = 100 (Obj. & Gap) | JSSP $15 \times 15$ (Obj. 6 Gap) |
> > > > | --- | --- | --- |
> > > > | GAZ PTP GT | 8.06 $\pm$ 0.04 & 3.8% $\pm$ 0.5% | 1505.2 $\pm$ 55.7 & 22.0% $\pm$ 3.8% |
> > > > | GAZ Single Vanilla | 10.76 $\pm$ 0.04 & 38.6% $\pm$ 0.5% | 2823.6 $\pm$ 1787.5 & 129.8% $\pm$ 145.4%
> > > > | GAZ Single TB Greedy | 10.30 $\pm$ 0.19 & 32.6% $\pm$ 2.5% | 1532.5 $\pm$ 20.3 & 24.7% $\pm$ 1.6%
> > > > | GAZ Single TB Sampled | 10.82 $\pm$ 0.20 & 39.4% $\pm$ 2.6% | 1910.8 $\pm$ 498.3 & 55.5% $\pm$ 40.6%

---

> > ### Comment · Reviewer_T4pL · 2022-11-17
> > **Thanks for detailed response**
> >
> > Thanks for your detailed response. I think it has helped me a lot to build intuition for why the problem was modelled the way it was now.
> >
> > I would just like to remark one thing related to this: in the revision of the paper itself, the new content added to provide this extra intuition is rather later (largely in an appendix, and a new paragraph very late into the paper referring to the appendix). I absolutely agree with the majority of this stuff (explanations of the alternative algorithms, extra results, etc. being in an appendix). However, if other readers of the paper are anything like me, I imagine they might also lack the intuition to fully follow along until they reach the end of the paper. A couple of brief sentences earlier one, maybe also explicitly pointing out the issue with inaccurate / difficult to estimate value functions, and how the 2-player models explicitly changes this difficulty, could possibly be helpful.
> >
> > I'm just providing that as a tip though, something that I think would have helped me personally. But if you believe it would be better to leave as is, I'm happy with that too (and certainly I don't want to demand more changes to the PDF right now at what I think is the end of the discussion phase?).

---

> > > ### Author Response · Authors · 2022-11-18
> > > **Thank you for the swift reply.**
> > >
> > > Thank you for the swift reply. We are happy that our response provided more intuition and are grateful that you raised your score to an 8.
> > > As there is still some time left, we gladly took your advice and added an extra sentence to the end of Section 3.2, "Motivation for the two-player game".
> > >
> > > Thank you again for your review and helping us improve our paper!

---

### Official Review · Reviewer_8mG6 · 2022-10-24

**Confidence:** 4
**Correctness:** 3
**Technical Novelty And Significance:** 3
**Empirical Novelty And Significance:** 3
**Recommendation:** 6

**Clarity, Quality, Novelty And Reproducibility:**

- Notation is confusing. Can $V^{\pi, \mu}(s_t, s_l’)$ be expressed as $V^{\pi}(s_t) - V^{\mu}(s_l’)$?  The same for other $Q$ and $A$. Thus, the proof of Lemma 1 (in Appendix) becomes straightforward.
- States $s$ are confused with $s(\zeta_1, \zeta_2)$.
- Why do we need to align $s_t$ and $s_l’$ in Figure 1?
- Many formula need to be marked as “(#)” for easier discussion.


**Strength And Weaknesses:**

**Strength:**

1. It is interesting to use self-competitive planning TSP or JSSP with binary episode reward (win or lose). This binary episode reward somehow handles the reward designing problem, which is an issue on RL problems.

**Weakness:**

1. This method trains from scratch for each problem size, compared to those GNN-DRL methods (cited in this paper) which are size-agnostic methods. This makes it inefficient since the problem size varies in such problems.

2. The performances for TA benchmarks 15x15, 20x20, 30x20 (for JSSP) are all worse than those obtained in Schedulenet (Park et al., 2022). Namely 16.6%, 22%, 32.3% (even with MCTS), w.r.t. schedulenet’s 15.3%, 17.2%, 18.7% respectively. Even, the bigger the problem sizes are, the worse the gap ratios are. The same for TSP. Thus, this makes this paper weaker or skeptical.

3. This paper mentions that MCTS method must need more inference time than DRL method in testing. However, the time is not included in the experiments.

4. The presentation is quite confusing. See the next section.









**Summary Of The Paper:**

This paper introduced Gumbel AlphaZero Play-to-Plan (GAZ PTP), a self-competitive method combining greedy rollouts as in self-critical training with the planning dynamics of two-player games.
The main idea is to compute a scalar baseline from the agent’s historical performances and to reshape an episode’s reward into a binary output, indicating whether the baseline has been exceeded or not.
GAZ PTP leverages the idea of self-competition and directly incorporates a historical policy into the planning process instead of its scalar performance. Based on the Gumbel AlphaZero (GAZ), the agent learns to find strong trajectories by planning against possible strategies of its past self.
Then, experiments on the Travel Salesman Problem (TSP) and Job-Shop Scheduling Problem (JSSP) confirm that the method learns strong policies in the original task with a low number of search simulations.


**Summary Of The Review:**

It is interesting to use self-competitive planning TSP or JSSP with binary episode reward (win or lose). However, when compared to other DRL methods, the performances of this paper are not so good. The trained model is not size-agnostic, and thus a big weakness of it. In summary, we tend to reject this paper.

---

> ### Author Response · Authors · 2022-11-16
> **Response to Reviewer 8mG6 (Part 1)**
>
> Dear reviewer,
>
> Thank you for reviewing our paper and for the constructive feedback! We would like to start by addressing your comments regarding *Clarity, Quality, Novelty And Reproducibility*, and then discuss the *Weaknesses* raised. We indicate changes made to the manuscript in **bold**.
>
> ### Clarity, Quality, Novelty And Reproducibility
>
> > Notation is confusing. Can $V^{\pi, \mu}(s_t,s_l')$  be expressed as $V^\pi(s_t)−V^μ(s_l')$? The same for other Q and A. Thus, the proof of Lemma 1 (in Appendix) becomes straightforward.
>
> We introduced the notation $V^{\pi, \mu}$ to indicate that in our self-competitive formulation, this is the quantity of interest that takes over the role of $V^{\pi}$. In particular, the value network is trained to approximate $V^{\pi, \mu}$ (instead of $V^{\pi}$, as usual). The action-selection mechanism of GAZ then stays the same; only the terms $V^{\pi}, Q^{\pi}$ and $A^{\pi}$ are swapped with $V^{\pi, \mu}, Q^{\pi, \mu}$ and $A^{\pi, \mu}$, which is why the notation is convenient. **To clarify this and make this change more explicit, we added the proposed altered logit update in equation (7) in Section 3.2.**
>
> As to your question, you are right, $V^{\pi, \mu} = V^\pi - V^\mu$, because the policies are independent of each other (see eq. (10) in the proof of Lemma 1). As you mentioned, the proof is indeed straightforward. There is no surprise, as the setup (value function depends on both states, policy not) is in fact designed to work this way. This is important because if we allowed  the policy net to also glimpse at the states encountered by the historical policy (as in the value network), the propagation of GAZ’s policy improvement would no longer be guaranteed.
>
> > States $s$ are confused with $s(\zeta_1,\zeta_2)$.
>
> Thank you for pointing this out. We agree that the notation $s(\zeta_1, \zeta_2)$ is confusing.  **For this reason, we changed to the regular $\text{sgn}$-function notation.**
>
> > Why do we need to align $s_t$ and $s_l'$ in Figure 1?
>
> (Remark: Please note that Figure 1 is now Figure 2 in Appendix B.2)
> One of the paper’s goals is to avoid losing information about intermediate states of the historical policy’s rollout – something that happens in both self-critical and self-competitive methods when the agent’s historical performance is summarized as a single scalar. This goal is achieved by comparing states encountered by the learning actor with states of the greedy actor at *similar timesteps*  (i.e., the learning actor sees how well it is doing by seeing how the old version of itself is doing in a *comparable* situation). This is the reason for the ‘move-by-move’ setup and thus, $s_t$ and $s_l’$ are aligned.
>
> For an in-depth discussion of why comparing pairs of states is sensible in situations where predicting the expected outcome of an episode is hard, please see our response to Reviewer T4pL below and the newly added Section “D. Baselining timesteps with value estimates” in the appendix.
>
> > Many formula need to be marked as “(#)” for easier discussion.
>
> Thanks for this suggestion. We initially tried to limit the numbering to equations referenced in the text but **added additional numbering. We especially numbered the lines in the “A. Proofs” section in the appendix.**

---

> > ### Author Response · Authors · 2022-11-16
> > **Response to Reviewer 8mG6 (Part 2)**
> >
> > ### Weaknesses
> >
> > General remark: We would like to emphasize that the main contribution of our paper consists of inserting the concepts of self-critical training into the policy improvement mechanisms of GAZ via the proposed self-competitive framework. We do not aim to present a new learned state-of-the-art solver for certain combinatorial optimization problems (COPs). Thus, the focus of our experiments lies in the comparison with different GAZ variants. We chose COPs due to several reasons (copied from response to Reviewer vDBJ above):
> >
> > a.) Policy gradient methods using self-critical training – which serve as a motivation for our approach – have shown great success in several classes of COPs (with [1] being the most prominent example mentioned in the paper).
> >
> > b.) They provide classes for which the size (and environment) of the problem can be increased easily, and the difficulty increases quickly. This is convenient for analyzing how the optimality gap evolves in different variants of GAZ. In particular, in the self-competitive formulation, we work with the assumption that the expected final outcome of an episode can be hard to predict. This becomes more and more relevant for larger problem sizes. As you mentioned, the equal episode length makes the method’s analysis more consistent.
> >
> > c.) In many COPs, it is hard to improve the performance at the end of the training to “squeeze out” the remaining percentages in the optimality gap. This is especially true for TSP, where, for the considered sizes, simple heuristics already yield good solutions, but more sophisticated strategies need to be found to adapt to individual instances. In later training stages, even subtle errors in predicting the expected outcome can stop the agent from improving, which makes it suitable for testing our hypotheses. This 'squeezing out the last percentages' is a little less relevant for JSSP, as the optimality gaps for good heuristics are higher (as for example reported in [2]).
> >
> > The results of [1] and [2] in Table 1 are only listed in perspective, that is why a deeper comparison is not necessary. As you correctly point out, they do not reflect the current state-of-the-art of DRL methods for COPs. We agree that this might not have been communicated clearly in the paper. **Thus, we added a paragraph at the beginning of Section “5.3 Results” to clarify why we chose [1] and [2].**
> >
> > > This method trains from scratch for each problem size, compared to those GNN-DRL methods (cited in this paper) which are size-agnostic methods. This makes it inefficient since the problem size varies in such problems.  [...] The trained model is not size-agnostic, and thus a big weakness of it.
> >
> > *Size agnosticism*: This is only partly correct. Our transformer model for TSP is entirely size-agnostic (and structurally very similar to the model used in [1], but tours are not created auto-regressively). The model for JSSP consists of two stacked Transformers. It is agnostic to the *number of jobs* but not the *number of machines*, as machines are identified by learnable position embeddings (and the number of position embeddings must be specified). We chose the architecture for TSP due to its similarity to [1] (which uses self-critical training and motivates our method). The design of our architecture for JSSP is structurally similar to the TSP model (especially the pointing mechanism for the policy), which we deem convenient for a consistent evaluation.
> >
> > We fully agree that the architecture for JSSP in particular is not an optimal choice. A GNN exploiting the disjunctive graph representation is more suited when the goal is a fast and optimized learned solver which needs to generalize to unseen sizes. However, as pointed out above, we focus on comparing the different GAZ variants. More importantly, even with a suboptimal model architecture, the results are valid because we use an almost identical architecture across the compared GAZ variants (not fully identical due to the different inputs to the network heads).

---

> > > ### Author Response · Authors · 2022-11-16
> > > **Response to Reviewer 8mG6 (Part 3)**
> > >
> > > > The performances for TA benchmarks 15x15, 20x20, 30x20 (for JSSP) are all worse than those obtained in Schedulenet (Park et al., 2022). Namely 16.6%, 22%, 32.3% (even with MCTS), w.r.t. schedulenet’s 15.3%, 17.2%, 18.7% respectively. Even, the bigger the problem sizes are, the worse the gap ratios are. The same for TSP. Thus, this makes this paper weaker or skeptical.
> > >
> > > Our comments above apply, but we would furthermore like to notify you of the added experiments for our method’s variant GAZ PTP GT (where also in the tree search, the greedy actor’s moves are chosen greedily), which yield improved results for TSP (and are comparable for JSSP).
> > >
> > > > This paper mentions that MCTS method must need more inference time than DRL method in testing. However, the time is not included in the experiments.
> > >
> > > We agree that this was communicated ambiguously in the manuscript. **We updated the paragraph “Experimental goal” in Section 5** (and especially removed the word “always” in the mention of MCTS’s runtime). MCTS is slower because it adds an additional network evaluation per search simulation and move (in contrast to ‘simply’ rolling out a policy network once, which needs exactly one network evaluation per move).
> > > Since our aim was not to present a new problem-specific learned solver, we do not see a benefit in  reporting runtimes of the greedy policy rollouts for comparison with [1] and [2]. Our main goal is to show the performance of GAZ PTP compared with applying GAZ in a single-player way.  We think that it will not lead to additional insights about our method and thus decided to not report runtimes.
> > >
> > > We want to thank you for reading our paper and for the valuable feedback, which helped us a lot to refine the manuscript! Please let us know if there are additional questions.
> > >
> > > ---
> > >
> > > [1] Wouter Kool, Herke Van Hoof, and Max Welling. Attention, learn to solve routing problems! International Conference on Learning Representations, 2018.
> > > [2] Cong Zhang, Wen Song, Zhiguang Cao, Jie Zhang, Puay S. Tan, and Xu Chi. Learning to dispatch for job shop scheduling via deep reinforcement learning. Advances in Neural Information Processing Systems, 2020.

---

> > > > ### Comment · Reviewer_8mG6 · 2022-11-26
> > > > **Thanks for the reply**
> > > >
> > > > Thanks for the reply which addressed most of my concerns. I raise my grade to "above". It is still expected to have SOTA, though.

---

### Official Review · Reviewer_vDBJ · 2022-10-24

**Confidence:** 5
**Correctness:** 4
**Technical Novelty And Significance:** 3
**Empirical Novelty And Significance:** 2
**Recommendation:** 8

**Clarity, Quality, Novelty And Reproducibility:**

The paper is well written. The proposed approach is novel and interesting. There are sufficient details provided for the  experiments, but the results might not be fully reproducible.


**Strength And Weaknesses:**

The idea of competing against a playout of a historical variant is interesting, and it is inserted in the AlphaZero framework in an elegant way.

The approach proposed in the paper replaces the problem of state-value estimation with a pairwise estimation problem. Pairwise comparison were tested previously in two-player game frameworks as well (with moderate success), but it is unclear that pairwise approaches are less frequent because of the particular approaches or because they proved more difficult than the original estimation problem in the test domains. Nevertheless the approach proposed is interesting, and there will always be domains that are more suited to pairwise comparisons.

An apparent limitation, which does not seem to be highlighted sufficiently in the paper, is that length of the episodes need to be equal. This is not a problem in the test domains considered in the paper, but many single agent problems do have varying episode length.

Using combinatorial optimization problems as test domains is somewhat surprising choice, but maybe that was due to the necessity of equal episode length and the efficiency of the pairwise comparisons.


**Summary Of The Paper:**

The paper proposes an MCTS/AlphaZero-type algorithm for single player task. There several crucial deviation from the standard variant: it is using the Gumbel variant of AlphaZero, and more crucially, instead of estimating the state-value function, it is biasing it (adversarily) with returns obtained in playouts using a historical version of the player. The algorithm is applied on TSP and JSSP instances obtaining strong performances.


**Summary Of The Review:**

The paper proposes some interesting variation of state-of-the-art algorithms, which may be successful in some domain. The applicability is somewhat limited by the requirement of uniform length episodes, and by the suitability of pairwise comparisons in the domain, but there likely exist sufficient number of domains were the approach is relevant.

---

> ### Author Response · Authors · 2022-11-16
> **Response to Reviewer vDBJ**
>
> Dear reviewer,
>
> We sincerely thank you for taking the time to read and review our paper! We very much appreciate that you find our proposed method novel and interesting and the paper well-written. In the following, we would like to address your points (we indicate changes made to the manuscript in **bold**):
>
> > An apparent limitation, which does not seem to be highlighted sufficiently in the paper, is that length of the episodes need to be equal. This is not a problem in the test domains considered in the paper, but many single agent problems do have varying episode length.
>
> Thank you for pointing this out. Indeed, in this paper, we only consider problem classes with constant episode length. However, the ‘game-playing’ method is easily straightforwardly extended to varying episode lengths: If the greedy actor finishes the episode first, we only consider its terminal state in all the nodes of the learning actor’s subsequent MCTS (i.e., the learning actor keeps its turn until the end and compares its state after an action constantly to the greedy actor’s terminal state). On the other hand, if the learning actor finishes first, nothing special has to be considered: we compare its trajectory to the greedy rollout of the greedy actor as usual to obtain the binary reward. **To highlight this possible extension we added a note in the paragraph ‘Limitations’, Section 5.3.**
> As you pointed out, there are domains for which this pairwise state comparison is more suited than others.
>
> > Using combinatorial optimization problems as test domains is somewhat surprising choice, but maybe that was due to the necessity of equal episode length and the efficiency of the pairwise comparisons.
>
> We chose combinatorial optimization problems (COPs) due to several reasons:
>
> a.) Policy gradient methods using self-critical training – which serve as a motivation for our approach – have shown great success in several classes of COPs (with [1] being the most prominent example mentioned in the paper).
>
> b.) They provide classes for which the size (and environment) of the problem can be increased easily, and the difficulty increases quickly. This is convenient for analyzing how the optimality gap evolves in different variants of GAZ. In particular, in the self-competitive formulation, we work with the assumption that the expected final outcome of an episode can be hard to predict. This becomes more and more relevant for larger problem sizes. As you mentioned, the equal episode length makes the method’s analysis more consistent.
>
> c.) In many COPs, it is hard to improve the performance at the end of the training to “squeeze out” the remaining percentages in the optimality gap. This is especially true for TSP, where, for the considered sizes, simple heuristics already yield good solutions, but more sophisticated strategies need to be found to adapt to individual instances. In later training stages, even subtle errors in predicting the expected outcome can stop the agent from improving, which makes it suitable for testing our hypotheses. This 'squeezing out the last percentages' is a little less relevant for JSSP, as the optimality gaps for good heuristics are higher (as for example reported in [2]).
>
> > There are sufficient details provided for the experiments, but the results might not be fully reproducible.
>
> To ensure reproducibility of our work, we posted a link to an anonymous repository with all our implementations when the discussion forum opened (please see the comment above): https://anonymous.4open.science/r/GAZ-PTP-2A0F
> After the review process, the code will be made publicly available on GitHub.
>
> We would like to thank you for reading our paper and for the valuable feedback, which helped us a lot to refine our paper! Please let us know if there are additional questions.
>
> ---
>
> [1] Wouter Kool, Herke Van Hoof, and Max Welling. Attention, learn to solve routing problems! International Conference on Learning Representations, 2018.
> [2] Cong Zhang, Wen Song, Zhiguang Cao, Jie Zhang, Puay S. Tan, and Xu Chi. Learning to dispatch for job shop scheduling via deep reinforcement learning. Advances in Neural Information Processing Systems, 2020.

---

### Author Response · Authors · 2022-11-16
**General response to all reviewers: Summary of changes**

Dear reviewers,

Thank you all for the thoughtful and insightful reviews. We revised the paper accordingly to your feedback and highlighted the changes in the manuscript in blue. We summarize our changes as follows (we added the alias of the respective reviewer in square brackets to indicate the individual response for further details):

- [vDBj] We added a note regarding varying episode lengths in the paragraph “Limitations” (p. 9).
- [8mG6] The proposed logit update is made explicit as eq. (7) in the motivation for the two-player game (p. 4).
- [8mG6] We rephrased the experimental goal (p. 7). We added a note that our focus lies on the comparison with various GAZ variants, and that we do not present a new state-of-the-art solver for TSP/JSSP.
- [8mG6] We added a note at the beginning of Section 5.3 on why we list the results of [1] and [2], even though they do not reflect the current state-of-the-art in DRL methods for TSP/JSSP (p. 8).
- [T4pL] We now present two variants of our proposed method: GAZ PTP **ST**, where in the MCTS the greedy actor’s actions are **sampled** (as before), and GAZ PTP **GT**, where in the MCTS the greedy actor’s actions are chosen **greedily**.
    - Introduction of both variants with pros and cons is summarized in paragraph “(i) Choice of actions” (Section 4.2, p. 6).
    - Main results of GAZ PTP GT are added to Table 1 (p. 8) and curves for seed experiments are added to Figure 1 (former Figure 2) (p.9).
    - GAZ PTP GT is also included in the results discussion on pp. 8-9.
    - In addition, we added a short section “B.6 Efficient implementation of GAZ PTP GT” to the appendix (p. 16).
- [T4pL] We added an entirely new section “D. Value estimates as timestep-dependent baselines” to the appendix (pp. 22-23). In the main text, it is referred to in the paragraph “Value estimates as baselines” on p. 9.
The section explores the sensibility of comparing pairs of states in a value function and shows additional experiments comparing this notion with baselining the Q-values with values estimates.

**Minor fixes**:

- [8mG6] Changed notation for sign of accumulated reward difference (p. 4)
- [8mG6] Added additional equation numbering in main text and numbered lines in the proofs.
- [T4pL] Fixed references.

**Additional minor fixes (unrelated to the reviews):**

- Figure 1 was moved to Figure 2 (p. 15) in Appendix B.
- Table 1 (p. 8): The main results for seed 42 of **GAZ Single N-Step** changed, as we had to rerun the seed 42 experiments due to an error in the config file. (The updated results do not change the story of the paper.)
- Appendix C.: Fixed a typo in the hyperparameters on p. 21, added additional sentence about the value head (p. 18), added additional environmental detail for JSSP (p. 20).

---

### Decision · Program_Chairs · 2023-01-20

**Decision:**

Accept: poster

**Justification For Why Not Higher Score:**

While the method in the paper is interesting and shows good results compared to other AlphaZero baselines, as one of the reviewers pointed out, it does not achieve SoTA performance on the domains studied. So while this paper presents a useful improvement of the AZ framework, its impact is somewhat limited by not also outperforming other classes of algorithms.

**Justification For Why Not Lower Score:**

The idea is interesting and shows how to improve AZ's performance in particularly challenging domains like combinatorial optimization. I expect it will be impactful for people working in planning and model-based RL and is worth disseminating to the community.

**Metareview: Summary, Strengths And Weaknesses:**

This paper introduces a new method for training AlphaZero-style algorithms, building upon the work of Gumble AlphaZero (GAZ), by incorporating the idea of self-competition. The main idea is to compute two trajectories using two different actors: one which selects actions greedily, and one which selects actions using MCTS (by planning out future trajectories of both actors, in order to improve over the greedy actor). The value function, rather than learning state values, learns pairwise values (comparing the strength of the two players). The method is evaluated on combinatorial optimization problems, and is shown to outperform other AlphaZero-based baselines.

All the reviewers cited the interestingness and novelty of the approach, with one reviewer emphasizing that it is an elegant addition to the AlphaZero framework. In the original reviews, all the reviewers pointed out some issues with clarity, which were sufficiently addressed by the authors during the rebuttal. As such, I feel this is a strong paper and the method is worth sharing at ICLR. I recommend acceptance.

**Note From Pc:**

if the above contains the word "oral" or "spotlight" please see: "oral" presentation means -> notable-top-5% and "spotlight" means -> notable-top-25%. As stated in our emails, we are disassociating presentation type from AC recommendations

**Summary Of Ac-Reviewer Meeting:**

N/A